# MaskPro: Linear-Space Probabilistic Learning for Strict (N:M)-Sparsity on LLMs

**Yan Sun**[*]
School of Computer Science
Faculty of Engineering
The University of Sydney
`ysun9899@uni.sydney.edu.au`

**Qixin Zhang**[*]
Generative AI Lab
College of Computing and Data Science
Nanyang Technological University
`qixin.zhang@ntu.edu.sg`

**Zhiyuan Yu**
University of Science and Technology of China
`yuzhiyuan@mail.ustc.edu.cn`

**Xikun Zhang**
RMIT University
`xikun.zhang@rmit.edu.au`

**Li Shen**
School of Cyber Science and Technology
Shenzhen Campus of Sun Yat-sen University
`mathshenli@gmail.com`

**Dacheng Tao** [†]
Generative AI Lab
College of Computing and Data Science
Nanyang Technological University
`dacheng.tao@ntu.edu.sg`

## Abstract

The rapid scaling of large language models (LLMs) has made inference efficiency a primary bottleneck in the practical deployment. To address this, semi-structured sparsity offers a promising solution by strategically retaining $N$ elements out of every $M$ weights, thereby enabling hardware-friendly acceleration and reduced memory. However, existing (N:M)-compatible approaches typically fall into two categories: rule-based layerwise greedy search, which suffers from considerable errors, and gradient-driven combinatorial learning, which incurs prohibitive training costs. To tackle these challenges, we propose a novel linear-space probabilistic framework named MaskPro, which aims to learn a prior categorical distribution for every $M$ consecutive weights and subsequently leverages this distribution to generate the (N:M)-sparsity throughout an $N$-way sampling without replacement. Furthermore, to mitigate the training instability induced by the high variance of policy gradients in the super large combinatorial space, we propose a novel update method by introducing a moving average tracker of loss residuals instead of vanilla loss. Finally, we conduct comprehensive theoretical analysis and extensive experiments to validate the superior performance of MaskPro, as well as its excellent scalability in memory efficiency and exceptional robustness to data samples. Our code is available at `https://github.com/woodenchild95/Maskpro.git`.

## 1 Introduction

Recent studies have witnessed the rapid advancement of LLMs across various domains, establishing them as a highly promising solution for a wide range of downstream tasks (Hendrycks et al., 2020; Brown et al., 2020; Achiam et al., 2023). However, the massive parameter size introduces significant overhead in both training and inference (Touvron et al., 2023; Grattafiori et al., 2024), underscoring the pressing need for efficient approaches in real-world applications (Shen et al., 2023; Zhou et al., 2024). In response, semi-structured sparsity has emerged as a technique with considerable practical potential, as its acceleration can be efficiently harnessed by accelerators (Mishra et al., 2021; Pool et al., 2021; Fan et al., 2025b;a). Specifically, it adopts a designated sparsity pattern, retaining only

---

[*]These authors contributed equally to this work.
[†]Corresponding author. Dr. Tao's research is partially supported by NTU RSR and Start Up Grants.

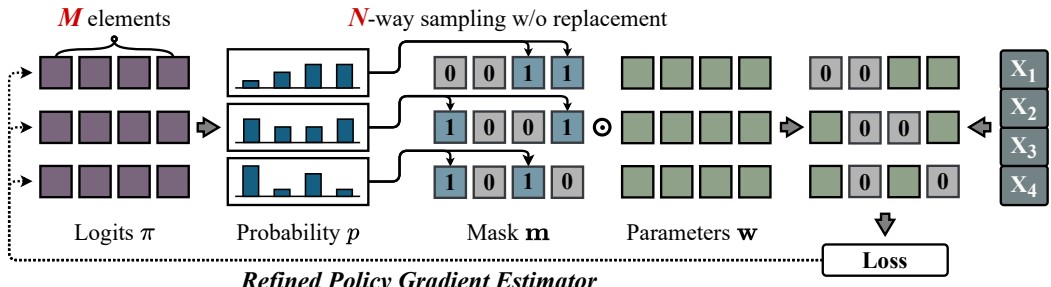

Figure 1: Implementation of our proposed MaskPro for learning (2:4)-sparse masks.

$N$ out of every $M$ consecutive weights, a scheme commonly referred to as (N:M)-sparsity. Owing to its effective support from parallel computing libraries, its inference performance is exceptionally efficient, offering a viable path toward the practical and scalable local deployment of LLMs.

Although its procedural design is relatively straightforward, effectively implementing (N:M)-sparsity while preserving model performance still remains a formidable challenge. One major obstacle lies in its enormous combinatorial scale, making it extremely difficult to identify the optimal mask. Existing methods can be broadly classified into two main branches. The first category encompasses rule-based approaches that bypass backpropagation by leveraging a calibration set to greedily minimize layerwise errors through the objective $\min_{\mathbf{m}} \|\mathbf{wx} - (\mathbf{m} \odot \mathbf{w})\mathbf{x}\|^2$ (Frantar & Alistarh, 2023). Based on this, a series of variants incorporating auxiliary information, e.g., $l_2$-norm of input activations (Sun et al., 2023) and gradients (Das et al., 2023; Dong et al., 2024) have been further applied, leading to certain improvements. However, such handcrafted metrics inherently suffer from considerable gaps with the end-to-end loss, ultimately capping the potential effectiveness of these methods. To address this, Fang et al. (2024) propose a learning-based method MaskLLM. Specifically, it determines the optimal solution by directly optimizing the objective $\min_{\mathbf{m}} f(\mathbf{m} \odot \mathbf{w})$ in generation tasks on a large dataset. MaskLLM achieves remarkable results, but its training costs are prohibitively high, even exceeding the overhead of finetuning the LLM itself. For instance, training the (N:M)-sparsity on $d$-dimensional weights requires at least additional $\mathcal{O}\left(\binom{M}{N}\frac{d}{M}\right)$ memory to save the logits. As $N$ and $M$ scale up, this memory overhead can even grow exponentially, yielding extremely poor scalability.

**Our Motivation.** Existing solutions either suffer from inherent biases or incur prohibitively high training costs, making them difficult to implement. This motivates us to further explore a memory-efficient learning-based method for this problem. Naturally, probabilistic modeling combined with efficient policy gradient estimators (PGE) emerges as a promising study. However, due to the vast combinatorial space and large model size, the variance of policy gradients can become so substantial that training is nearly impossible. Moreover, the memory overhead required to store the logits remains excessively large. To enable effective training, these two challenges must be adequately addressed.

To tackle these challenges, we introduce a linear-space probabilistic framework termed as MaskPro. Compared with the current state-of-the-art MaskLLM (Fang et al., 2024), instead of the probability distributions for all possible masks of $M$ weights, our proposed MaskPro establishes a categorical distribution for every $M$ consecutive elements and then utilizes this distribution to generate the (N:M)-sparsity through an $N$-way sampling without replacement. This implies that for any (N:M)-sparsity pattern, we only require $\mathcal{O}(d)$ memory to store the logits. Furthermore, we propose a novel PGE update to accelerate and stabilize the entire training process, which modifies the independent loss metric in vanilla PGE by the loss residuals with a moving average tracker. We provide the rigorous theoretical analysis for our probabilistic modeling and prove the unbiasedness and variance reduction properties of the proposed PGE. To investigate its effectiveness, we conduct extensive experiments on several LLMs and report the performance across various downstream tasks. Experiments indicate that the proposed MaskPro can achieve significant performance improvements while maintaining memory usage comparable to rule-based methods, with substantially lower training overhead than MaskLLM. Moreover, the MaskPro method demonstrates remarkable robustness to data samples, which can achieve stable performance even with **only 1 training sample**.

We summarize the main contributions of this work as follows:

- We propose a linear-space probabilistic framework MaskPro, formulating the (N:M)-sparsity as a process of $N$-way samplings without replacement within a categorical distribution over $M$ consecutive elements, which reduces the memory for logits from $\mathcal{O}\left(\binom{M}{N}\frac{d}{M}\right)$ to $\mathcal{O}(d)$.

- We propose an enhanced policy gradient that substitutes the raw loss in standard policy gradients with per-minibatch loss residuals. To maintain stability, we further incorporate a moving-average baseline that adaptively tracks the residual dynamics during training.

- We provide the comprehensive theoretical analysis to understand the memory effectiveness of MaskPro and the variance reduction properties of the proposed policy gradient update. Extensive experiments validate its significant performance. Moreover, it exhibits outstanding robustness to data samples, maintaining stable results even with only 1 training sample.

## 2 RELATED WORK

**Model Pruning.** Model pruning is an important compression technique that has been adopted in several domains (Han et al., 2015; Frankle & Carbin, 2018; Liu et al., 2019; Xia et al., 2023b; Sun et al., 2023; Sreenivas et al., 2024; Luo et al., 2025). It also demonstrates strong practicality in real-world applications of LLMs. A series of structured learning and optimization methods on pruning and training have been proposed and widely applied, including the depth- and width-based (Ko et al., 2023), kernel-based (Xia et al., 2023a), LoRA-based (Chen et al., 2023; Zhang et al., 2023; Zhao et al., 2024), row- and column-based (Ashkboos et al., 2024), channel-based (Gao et al., 2024b; Dery et al., 2024), layer-based (Yin et al., 2023; Men et al., 2024; Zhang et al., 2024a), attention head-base (Ma et al., 2023), MoE-based (Chen et al., 2022; Xie et al., 2024). These methods leverage a prune-train process to effectively reduce the number of effective parameters while maintaining efficient training, bring a promising solution for the practical application and deployment of LLMs in the real-world scenarios. However, structured pruning typically considers a specific model structure as the minimal pruning unit, which can significantly impact the model's performance. The fundamental unit of a model is each individual weight, implying that unstructured pruning methods generally have higher potential on the performance (Frantar & Alistarh, 2023; Jaiswal et al., 2023). Such methods can typically identify a fine-grained mask that closely approaches the performance of dense models.

**Semi-structure Pruning.** Due to the inability of GPUs and parallel computing devices to perfectly support arbitrary element-wise sparse computations, the practical efficiency of sparse models remains significantly constrained. Semi-structured sparsity offers a promising pathway for practical applications (Zhou et al., 2021; Zhang et al., 2022; Lu et al., 2023), which is also called (N:M)-sparsity. A series of methods supporting semi-structured sparsity have been consistently applied, primarily including rule-based (Han et al., 2015; Frantar & Alistarh, 2023; Sun et al., 2023; Das et al., 2023; Dong et al., 2024; Zhang et al., 2024b) and learning-based (Holmes et al., 2021; Fang et al., 2024; Huang et al., 2025) approaches. Our work is the first to adopt policy gradients for learning semi-structured masks on LLMs. Enormous variance of policy gradients caused by the vast combinatorial space makes learning (N:M)-sparsity via PGE more challenging than those gradient-based methods.

## 3 PRELIMINARY

### 3.1 SEMI-STRUCTURED SPARSITY

The core idea of semi-structured sparsity aims to divide the entire weights $\mathbf{w} \in \mathbb{R}^d$ into groups of $M$ consecutive elements and then retain $N$ effective weights for each group. More specifically, we can formulate the semi-structured sparsity as the following combinatorial optimization problem:

$$\mathbf{m}^\star = \underset{\mathbf{m}=\{\mathbf{m}_i|\mathbf{m}_i \in \mathcal{S}^{N:M}\}}{\arg\min} \mathbb{E}_{\xi \sim \mathcal{D}}\left[f(\mathbf{m} \odot \mathbf{w}, \xi)\right], \tag{1}$$

where $f(\cdot)$ denotes the corresponding loss function, the symbol $\odot$ stands for the element-wise multiplication, $\xi \sim \mathcal{D}$ represents the minibatch sampled from the underlying distribution $\mathcal{D}$ and $\mathcal{S}^{N:M} = \left\{\mathbf{m}_i \in \mathbb{B}^{1 \times M} : \|\mathbf{m}_i\|_1 = N\right\}$ ($\mathbb{B}$ is the Boolean set and $\|\cdot\|_1$ denotes $l_1$ norm).

Generally speaking, in order to find the optimal mask $\mathbf{m}^\star$ for problem 1, we are confronted with two significant challenges: ***i) Huge Search Space:*** In the context of LLMs, the model parameter

scale $d$ can become extremely large, which will result in the search space for problem 1 reaching an astounding size of $\binom{M}{N}^{d/M}$; **ii) Non-Differentiability of Mask Selection:** The discrete nature of problem 1 prevents us from utilizing the gradient-based methods such as SGD (Lan, 2020), conditional gradient (Braun et al., 2022) and ZO Sun et al. (2025) to search for the optimal mask $\mathbf{m}^\star$.

To address these aforementioned issues, we will introduce an innovative probabilistic framework termed as MaskPro for problem 1 in the subsequent sections. Prior to that, we first review the state-of-the-art learning-based MaskLLM method (Fang et al., 2024).

## 3.2 Rethinking the General Probabilistic Modeling and its Inefficiency

Recent advance provides a learning method to address Problem 1, named MaskLLM (Fang et al., 2024). Specifically, for each group of $M$ consecutive weights, MaskLLM defines a categorical distribution with class probability $\left[p_1, p_2, \cdots, p_{|\mathcal{S}^{N:M}|}\right]$ where $\sum_i p_i = 1$, and each $p_i$ represents the probability of the corresponding element in $\mathcal{S}^{N:M}$. By random sampling, if a certain mask performs better, it is reasonable to increase the probability of the sampled mask. Otherwise, the sampling probability should be decreased. Thus, Problem 1 can be transformed as,

$$\{p^\star(\mathbf{m}_i)\} = \underset{\{p(\mathbf{m}_i)\}}{\arg\min} \, \mathbb{E}_{\xi \sim \mathcal{D}, \mathbf{m}=\{\mathbf{m}_i | \mathbf{m}_i \sim p(\mathbf{m}_i)\}} \left[ f(\mathbf{m} \odot \mathbf{w}, \xi) \right], \tag{2}$$

where $p(\mathbf{m}_i)$ is the categorical distribution of the $i$-th mask $\mathbf{m}_i$ over $\mathcal{S}^{N:M}$.

To enable the end-to-end training, MaskLLM further introduces Gumbel-Max(Gumbel, 1954) as reparameterization to relax the discrete sampling into a continuous form, making it naturally differentiable. This reparameterized loss-driven mask learning method is highly effective on various LLMs, providing a innovative perspective for addressing this problem.

However, the memory overhead in the MaskLLM training process is extremely large. Firstly, the backpropagation of gradients typically requires storing a large number of intermediate activation values and a substantial amount of optimizer states must be maintained during updates. A more notable issue is the separate probability assigned to each possible selection of $\mathbf{m}_i$ over $\mathcal{S}^{N:M}$, which may cause extreme memory explosion. Concretely, when learning (N:M)-sparsity for the weights $\mathbf{w} \in \mathbb{R}^d$, MaskLLM requires at least $\mathcal{O}\left(\binom{M}{N}\frac{d}{M}\right)$ space to save the logits for learning probabilities, which approximately reaches $\mathcal{O}\left(\frac{2^M}{M}d\right)$ at the worst case ($N \approx M/2$). This implies that the computational resources required by MaskLLM can even increase exponentially as $M$ becomes large, significantly limiting its scalability in practical scenarios, especially with extremely large model size.

## 4 Methodology

In this section, we present the details of our proposed MaskPro method. Specifically, in Section 4.1, we introduce the novel linear-space probabilistic framework to tackle the memory drawback of the vanilla sampling process in MaskLLM (Fang et al., 2024). Then, in Section 4.2, we propose to adopt the backpropagation-free policy gradient for training. Moreover, we further refine the logits update via utilizing the loss residual with a smoothing tracker instead of vanilla loss metric, which enhances the effectiveness and stability of the learning process.

## 4.1 MaskPro: A Linear-Space Probabilistic Relaxation

Before going into the details of our proposed MaskPro probabilistic framework, we first present a representation theory of the concerned N:M mask set $\mathcal{S}^{N:M} = \left\{\mathbf{m}_i \in \mathbb{B}^{1 \times M} : \|\mathbf{m}_i\|_1 = N\right\}$. In order to better illustrate our results, we need to introduce a new operation $\oplus$ for the coordinate-wise probabilistic sum of two vectors. Formally, for any $\mathbf{a} \in \mathbb{R}^{1 \times M}$ and $\mathbf{b} \in \mathbb{R}^{1 \times M}$, we define $\mathbf{a} \oplus \mathbf{b} = \mathbf{1}_M - (\mathbf{1}_M - \mathbf{a}) \odot (\mathbf{1}_M - \mathbf{b})$, where the symbol $\mathbf{1}_M$ denotes the $M$-dimensional vector whose all coordinates are 1. It is worth noting that this $\oplus$ is a symmetric associative operator, namely, $\mathbf{a} \oplus \mathbf{b} = \mathbf{b} \oplus \mathbf{a}$. Therefore, it also makes sense to apply the operation $\oplus$ to a set of vectors. Specifically, given multiple $M$-dimensional vectors $\{\mathbf{a}_1, \ldots, \mathbf{a}_N\}$, we can define that

$$\bigoplus_{i=1}^N \mathbf{a}_i = \mathbf{a}_1 \oplus \mathbf{a}_2 \oplus \cdots \oplus \mathbf{a}_N = \left( \mathbf{1}_M - \bigodot_{i=1}^N (\mathbf{1}_M - \mathbf{a}_i) \right). \tag{3}$$

With this operation $\oplus$, we then can derive a sparse representation for the N:M mask set $\mathcal{S}^{N:M}$, i.e.,

---

**Theorem 1 (Representation of N:M Sparsity)**

$$\mathcal{S}^{N:M} = \left\{ \bigoplus_{i=1}^{N} \mathbf{a}_i \; : \; \mathbf{a}_i \in \{\mathbf{e}_1, \dots, \mathbf{e}_M\}, \forall i \in [N] \; and \; \mathbf{a}_1 \neq \mathbf{a}_2 \neq \dots \neq \mathbf{a}_N \right\}, \quad (4)$$

where each $\mathbf{e}_j$ denotes the $j$-th basis vector of the space $\mathbb{R}^{1 \times M}$.

---

From a high-level viewpoint, Theorem 1 offers a parameter-reduced representation of the mask space $\mathcal{S}^{N:M}$. Notably, representing $N$ distinct $M$-dimensional vectors $\{\mathbf{a}_1, \dots, \mathbf{a}_N\}$ typically requires at most $(NM)$ unknown parameters. In contrast, the mask set $\mathcal{S}^{N:M}$ often has a enormous size of $\binom{M}{N}$. Particularly when $N$ is comparable to $M$, the parameter scale $NM$ of vectors $\{\mathbf{a}_1, \dots, \mathbf{a}_N\}$ can be significantly smaller than the space complexity $\binom{M}{N}$ of $\mathcal{S}^{N:M}$.

Motivated by the results of Theorem 1, if we represent each mask $\mathbf{m}_i \in \mathcal{S}^{N:M}$ in problem 1 as a probabilistic sum of $\{\mathbf{a}_{i,1}, \dots, \mathbf{a}_{i,N}\}$ where $\mathbf{a}_{i,j} \in \{\mathbf{e}_1, \dots, \mathbf{e}_M\}, \forall j \in [N]$ and $\mathbf{a}_{i,j_i} \neq \mathbf{a}_{i,j_2}, \forall j_1 \neq j_2$, then we naturally can reformulate our concerned mask selection problem 1 as a binary optimization with variables $\{\mathbf{a}_{i,j}\}_{j=1}^{N}, \forall i \in [\frac{d}{M}]$, that is to say,

$$\min_{\mathbf{a}_{i,j} \in \{\mathbf{e}_1, \dots, \mathbf{e}_M\}} \mathbb{E}_{\xi \sim \mathcal{D}} \left[ f\left( \bigoplus_{j=1}^{N} \mathbf{a}_{i,j} \odot \mathbf{w}_i, \xi \right) \right], \quad \text{s.t. } \mathbf{a}_{i,j_i} \neq \mathbf{a}_{i,j_2}, \forall j_1 \neq j_2 \in [N], \quad (5)$$

where the symbol $\mathbf{w}_i$ denotes the $i$-th group of the whole weight vector $\mathbf{w} \in \mathbb{R}^d$ and $i \in [\frac{d}{M}]$.

Notably, in Eq.5, we only employ $NM * \frac{d}{M} = Nd$ unknown parameters, which is significantly smaller than the $\left( \binom{M}{N} \frac{d}{M} \right)$ parameters scale used by the MaskLLM method. However, this new parameter-reduced formulation Eq.equation 5 of problem 1 still remains a discrete combinatorial optimization problem such that we cannot directly utilize gradient information to search for the optimal mask. To overcome this hurdle, we further introduce a novel probabilistic relaxation for problem 5 in the subsequent part of this section.

Note that in Eq.5, we restrict each group of variables $\{\mathbf{a}_{i,1}, \dots, \mathbf{a}_{i,N}\}$ to be $N$ distinct basis vectors in $\mathbb{R}^{1 \times M}$, that is, $\mathbf{a}_{i,j} \in \{\mathbf{e}_1, \dots, \mathbf{e}_M\}, \forall j \in [N]$ and $\mathbf{a}_{i,j_i} \neq \mathbf{a}_{i,j_2}, \forall j_1 \neq j_2 \in [N]$. In other words, we hope to identify an effective $N$-size subset from the basis vectors $\{\mathbf{e}_1, \dots, \mathbf{e}_M\}$, which closely resembles an $N$-way sampling-without-replacement process over $\{\mathbf{e}_1, \dots, \mathbf{e}_M\}$. Inspired by this finding, we design a novel continuous-relaxation framework named MaskPro for Eq.5, i.e., Firstly, we allocate a categorical distribution $\mathbf{p}_i = (p_{i,1}, \dots, p_{i,M})$ for each group of variables $\{\mathbf{a}_{i,1}, \dots, \mathbf{a}_{i,N}\}$. Subsequently, we employ every categorical distribution $\mathbf{p}_i$ to sequentially generate $N$ different random basis vectors $\{\mathbf{e}_{i,1}, \dots, \mathbf{e}_{i,N}\}$ throughout an $N$-way sampling-without-replacement trial where $\mathbf{e}_{i,j} \in \{\mathbf{e}_1, \dots, \mathbf{e}_M\}$ and $\mathbf{e}_{i,j_1} \neq \mathbf{e}_{i,j_j}, \forall j_1 \neq j_2$. Finally, we assign these sampled basis vectors to the variables $\{\mathbf{a}_{i,1}, \dots, \mathbf{a}_{i,N}\}$ by setting $\mathbf{a}_{i,j} := \mathbf{e}_{i,j}, \forall j \in [N]$.

Specifically, under the previously described probabilistic framework, the discrete problem equation 5 can naturally be converted into a continuous optimization task focused on learning the optimal categorical distributions $\mathbf{p}_i$ across the basis vectors $\{\mathbf{e}_1, \dots, \mathbf{e}_M\}$, that is,

$$\min_{\|\mathbf{p}_i\|_1 = 1, \forall i \in [\frac{d}{M}]} \Phi(\mathbf{p}) := \mathbb{E}_{\{\mathbf{a}_{i,j}\}_{j=1}^{N} \sim \mathbf{p}_i, \xi \sim \mathcal{D}} \left[ f\left( \bigoplus_{j=1}^{N} \mathbf{a}_{i,j} \odot \mathbf{w}_i, \xi \right) \right], \quad (6)$$

where $\{\mathbf{a}_{i,j}\}_{j=1}^{N} \sim \mathbf{p}_i$ represents the $N$-step sampling-without-replacement process guided by the categorical distribution $\mathbf{p}_i$. Note that representing all $\frac{d}{M}$ different categorical distributions $\{\mathbf{p}_i\}_{i=1}^{\frac{d}{M}}$ typically requires $\frac{d}{M} * M = d$ unknown parameters. Thus, by introducing randomness, the parameter scale of problem 6 can be further reduced from the previous $Nd$ of problem 5 to a linear $d$.

Next, we utilize the re-parameterization trick to eliminate the unit simplex constraint inherent in the problem 6, namely, $\{\mathbf{p}_i \in [0,1]^M : \|\mathbf{p}_i\|_1 = 1\}$. This step is crucial as it enables us to avoid the

computationally expensive projection operations. Specifically, we reset $\mathbf{p}_i := \mathrm{softmax}(\pi_i)$ where $\pi_i = (\pi_{i,1}, \ldots, \pi_{i,M})$ is the logits of softmax function. With this reformulation, we can transform the problem 6 as an unconstrained optimization regarding the logits $\pi := \{\pi_i\}_{i=1}^{\frac{d}{M}}$, that is,

$$\min_{\pi} \Phi(\pi) := \mathbb{E}_{\{\mathbf{a}_{i,j}\}_{j=1}^{N} \sim \mathrm{softmax}(\pi_i), \xi \sim \mathcal{D}} \left[ f\left( \bigoplus_{j=1}^{N} \mathbf{a}_{i,j} \odot \mathbf{w}_i, \xi \right) \right]. \tag{7}$$

To avoid repeatedly using the cumbersome notation $\bigoplus$, in the remainder of this paper, we define $\mathbf{m}_i := \bigoplus_{j=1}^{N} \mathbf{a}_{i,j}$ for any $i \in [\frac{d}{M}]$ and also use $p(\mathbf{m}_i|\pi_i)$ to denote the probability of our MaskPro generating the mask $\mathbf{m}_i$ under logits $\pi_i$. Then, the previous problem 7 can be rewritten as:

$$\min_{\pi} \Phi(\pi) := \mathbb{E}_{\xi \sim \mathcal{D}, \mathbf{m} = \{\mathbf{m}_i | \mathbf{m}_i \sim p(\mathbf{m}_i|\pi_i)\}} \left[ f\left( \mathbf{m} \odot \mathbf{w}, \xi \right) \right] = \int \mathbb{E}_{\xi} \left[ f\left( \mathbf{m} \odot \mathbf{w}, \xi \right) \right] p(\mathbf{m}|\pi) \mathrm{d}\mathbf{m}, \tag{8}$$

where $\mathbf{m} \in \mathbb{B}^{1 \times d}$ is the concatenation of all mask $\{\mathbf{m}_1, \ldots, \mathbf{m}_{\frac{d}{M}}\}$ and $p(\mathbf{m}|\pi) := \prod_{i=1}^{\frac{d}{M}} p(\mathbf{m}_i|\pi_i)$.

## 4.2 POLICY GRADIENT ESTIMATOR AND REFINED (N:M)-SPARSITY LEARNING

Thanks to the probabilistic formulation of Eq. 8, we thus can facilitate an efficient optimization via a policy gradient estimator. Specifically, we have the following equality:

$$\nabla \Phi(\pi) = \mathbb{E}_{\xi \sim \mathcal{D}, \mathbf{m} = \{\mathbf{m}_i | \mathbf{m}_i \sim p(\mathbf{m}_i|\pi_i)\}} \left[ f(\mathbf{m} \odot \mathbf{w}, \xi) \nabla \log \left( p(\mathbf{m}|\pi) \right) \right]. \tag{9}$$

As for the proof of Eq.9 and the specific calculation of $p(\mathbf{m}|\pi)$ in our MaskPro, please refer to Appendix C.2 and B. Note that Eq.9 can be computed purely with forward propagation. Therefore, we can update the logits variables $\pi$ via a mini-batch stochastic gradient descent, that is to say,

$$\pi_{t+1} = \pi_t - \eta f(\mathbf{m}_t \odot \mathbf{w}, \xi) \nabla \log \left( p(\mathbf{m}_t|\pi_t) \right). \tag{10}$$

Although Eq.(10) may perform well in elementary tasks, it faces one major challenge in the context of LLMs, which is caused by the inherent differences in loss values among different minibatch.

**Ambiguity on Mask $\mathbf{m}_t$ and Minibatch $\xi$.** The policy gradient updates logits based on the loss metric, aiming to encourage the logits to select masks that result in lower loss values. However, when the loss variation caused by mask sampling is significantly smaller than the loss variation caused by changing the minibatch, the loss metric alone cannot effectively distinguish whether the current mask is beneficial or detrimental. For example, we denote $\xi_{\mathrm{low}}$ as the minibatch whose loss is inherently low and $\xi_{\mathrm{high}}$ as the minibatch with high loss. Then we sample two masks and denote one that achieves lower loss by $\mathbf{m}_{\mathrm{good}}$ and the other by $\mathbf{m}_{\mathrm{bad}}$. There are typically two scenarios during training.

- $f(\mathbf{m}_{\mathrm{good}} \odot \mathbf{w}, \xi_{\mathrm{low}}) \leq f(\mathbf{m}_{\mathrm{bad}} \odot \mathbf{w}, \xi_{\mathrm{low}})$ and $f(\mathbf{m}_{\mathrm{good}} \odot \mathbf{w}, \xi_{\mathrm{high}}) \leq f(\mathbf{m}_{\mathrm{bad}} \odot \mathbf{w}, \xi_{\mathrm{high}})$.
- A bad case: $f(\mathbf{m}_{\mathrm{bad}} \odot \mathbf{w}, \xi_{\mathrm{low}}) \leq f(\mathbf{m}_{\mathrm{good}} \odot \mathbf{w}, \xi_{\mathrm{high}})$.

The first case is likely to hold in most cases, as a good mask can generally reduce the loss on most minibatches. But when the bad case occurs, Eq.(10) interprets that the lower-loss sample as the better one, yielding more erroneous learning on $m_{\mathrm{bad}}$. To better illustrate this phenomenon, we randomly select two minibatches during the training of LLaMA-2-7B and extract the logits at the 500-th iteration. We then sample 1000 masks and plot their *loss distributions*, as shown in Figure 2. It is clearly observed that $f(\mathbf{m}_{\mathrm{bad}} \odot \mathbf{w}, \xi_1) \leq f(\mathbf{m}_{\mathrm{good}} \odot \mathbf{w}, \xi_2)$. Such disparities between minibatches are quite common, causing Eq.(10) to frequently encounter conflicting information when learning solely based on loss value $f(\mathbf{m} \odot \mathbf{w}, \xi)$.

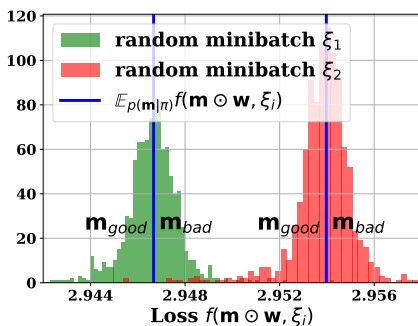

Figure 2: Loss-related misconceptions.

To address this issue, we propose to use the loss residual to update the logits, which can distinguish the loss variations independently caused by mask changes. By rethinking the first case above, to accurately evaluate whether a mask is better, we should fix the impact of minibatch. Similarly, we

---

**Algorithm 1** Learning (N:M)-Sparsity via MaskPro

---

**Input:** frozen weights $\mathbf{w}$, initial logits $\pi_0$, initial mask $\mathbf{m}_0$, learning rate $\eta$, smoothing coefficient $\alpha = 0.99$, smoothing tracker $\delta = 0$.

**Output:** learned logits $\pi_T$

1: **for** $t = 0, 1, 2, \cdots, T - 1$ **do**
2:     sample a minibatch $\xi$ for training
3:     reshape $\pi_t$ into groups of $M$ elements and calculate $p_t = \mathrm{softmax}(\pi_t)$ for each group
4:     perform $N$-way sampling without replacement by $p_t$ to generate the mask $\mathbf{m}_t$
5:     perform inference and calculate the loss residual $f(\mathbf{m}_t \odot \mathbf{w}, \xi) - f(\mathbf{m}_0 \odot \mathbf{w}, \xi)$
6:     update logits $\pi_{t+1} = \pi_t - \eta \left( f(\mathbf{m}_t \odot \mathbf{w}, \xi) - f(\mathbf{m}_0 \odot \mathbf{w}, \xi) - \delta \right) \nabla \log \left( p(\mathbf{m}_t | \pi_t) \right)$
7:     update the smoothing tracker $\delta = \alpha\delta + (1 - \alpha) \left( f(\mathbf{m}_t \odot \mathbf{w}, \xi) - f(\mathbf{m}_0 \odot \mathbf{w}, \xi) \right)$
8: **end for**

---

introduce $f(\mathbf{m}_t \odot \mathbf{w}, \xi) - f(\mathbf{m}_0 \odot \mathbf{w}, \xi)$ instead of $f(\mathbf{m}_t \odot \mathbf{w}, \xi)$ alone to evaluate whether the current sampled mask $\mathbf{m}_t$ is better than the baseline of initial $\mathbf{m}_0$. Thus, the update is refined as:

$$\pi_{t+1} = \pi_t - \eta \left( f(\mathbf{m}_t \odot \mathbf{w}, \xi) - f(\mathbf{m}_0 \odot \mathbf{w}, \xi) \right) \nabla \log \left( p(\mathbf{m}_t | \pi_t) \right). \tag{11}$$

In experiments, the effectiveness of Eq.(11) is significantly better than that of Eq.(10). However, it exhibits poor numerical stability. To further handle the potential numerical explosion during training, motivated by Zhao et al. (2011), we introduce a moving average tracker to evaluate the averaged loss residual under the current logits. Specifically, we reformulate Eq.(11) as follows:

$$\begin{aligned} \pi_{t+1} &= \pi_t - \eta \left( f(\mathbf{m}_t \odot \mathbf{w}, \xi) - f(\mathbf{m}_0 \odot \mathbf{w}, \xi) - \delta \right) \nabla \log \left( p(\mathbf{m}_t | \pi_t) \right), \\ \delta &= \alpha\delta + (1 - \alpha) \left( f(\mathbf{m}_t \odot \mathbf{w}, \xi) - f(\mathbf{m}_0 \odot \mathbf{w}, \xi) \right). \end{aligned} \tag{12}$$

Eq.(12) not only effectively distinguishes the loss variations caused by each sampled mask but also stabilizes its numerical distribution around zero through the $\delta$ term. This prevents aggressive logits updates caused by large loss variations, ensuring a more stable training process. We also provide a theoretical intuition and understanding for the $\delta$ term in Appendix C.3.3.

We summarize the training procedure in Algorithm 1. At $t$-th iteration, we first reshape the logits $\pi_t$ into groups of $M$ consecutive elements and then apply the softmax function to generate the corresponding probabilities $p_t$. Based on $p_t$, we perform an $N$-way sampling without replacement for each group, resulting in a strict (N:M)-sparse mask. We then calculate the policy gradient to update the current logits. By calculating the loss residual on the corresponding minibatch $\xi$, we can obtain the independent impact of the loss value. With the assistance of a smoothing tracker, we ensure that the distribution of loss residuals used for the policy gradient remains stable. Then we complete the policy gradient update of the logits. Finally, we update the smoothing tracker $\delta$. Regarding the final output, since the output consists of the logits $\pi_T$ of all weights, in our experiments, we directly select the top-$N$ positions with the highest logits within each group of $M$ elements as the mask. Actually, a more refined approach is to perform multiple $N$-way sampling-without-replacement processes and then evaluate them on a small calibration set to select the optimal mask.

## 5   UNBIASEDNESS AND VARIANCE REDUCTION

In this section, we primarily demonstrate the unbiasedness and variance-reduced properties of our proposed PGE update. For clarity of exposition, we denote these three updates as:

$$\begin{aligned} g_{\mathrm{p}} &= f(\mathbf{m} \odot \mathbf{w}, \xi) \nabla \log \left( p(\mathbf{m} | \pi) \right), \\ g_{\mathrm{r}} &= \left( f(\mathbf{m} \odot \mathbf{w}, \xi) - f(\mathbf{m}_0 \odot \mathbf{w}, \xi) \right) \nabla \log \left( p(\mathbf{m} | \pi) \right), \\ g_{\mathrm{sr}} &= \left( f(\mathbf{m} \odot \mathbf{w}, \xi) - f(\mathbf{m}_0 \odot \mathbf{w}, \xi) - \delta \right) \nabla \log \left( p(\mathbf{m} | \pi) \right), \end{aligned}$$

where $g_{\mathrm{p}}$ is the vanilla PGE, $g_{\mathrm{r}}$ is the update via loss residual and $g_{\mathrm{sr}}$ is the update via loss residual with smoothing tracker $\delta$. Then the following theorem holds (proof is deferred to Appendix C.3).

Table 1: Zero-shot evaluations of (2:4)-sparsity. In the test, we freeze weight updates and directly apply masks. The results corresponding to each model name reflects the evaluation of dense weights.

| | Wiki. | HellaS. | RACE | PIQA | WinoG. | ARC-E | ARC-C | OBQA | Memory |
|---|---|---|---|---|---|---|---|---|---|
| **GEMMA-7B** | 112.39 | 60.54 | 40.19 | 79.71 | 73.09 | 81.65 | 49.91 | 32.80 | — |
| - MASKLLM | — | 25.42 | 20.10 | 51.52 | 49.49 | 25.21 | 21.59 | 18.40 | 467.14 G |
| - MAGNITUDE | — | 25.23 | 21.24 | 51.85 | 50.75 | 26.43 | 21.84 | 12.40 | 16.32 G |
| - SPARSEGPT | — | 26.07 | 22.39 | 55.11 | 50.36 | 30.64 | 18.43 | 14.80 | 34.94 G |
| - WANDA | — | 26.80 | 22.78 | 56.47 | 48.86 | 32.66 | 17.75 | 13.60 | 29.63 G |
| - GBLM | — | 26.81 | 22.49 | 54.52 | 51.07 | 32.38 | 17.66 | 14.00 | 39.38 G |
| - PRUNER-ZERO | — | 25.27 | 21.63 | 53.21 | 50.75 | 24.58 | 22.70 | 15.20 | 39.38 G |
| - **MaskPro** | — | 26.97 | 23.26 | 57.88 | 52.82 | 32.92 | 22.65 | 16.40 | 48.63 G |
| **VICUNA-1.3-7B** | 11.86 | 56.32 | 41.91 | 77.37 | 69.46 | 74.28 | 42.41 | 34.60 | — |
| - MASKLLM | 14.91 | 49.07 | 39.13 | 75.24 | 65.35 | 65.57 | 33.57 | 25.60 | 331.16 G |
| - MAGNITUDE | 389.92 | 40.19 | 28.61 | 67.03 | 57.62 | 54.59 | 28.75 | 19.40 | 12.82 G |
| - SPARSEGPT | 24.93 | 44.87 | 37.81 | 70.62 | 63.30 | 62.92 | 32.42 | 25.00 | 22.20 G |
| - WANDA | 25.24 | 44.28 | 37.89 | 70.57 | 61.56 | 61.70 | 32.17 | 23.00 | 21.25 G |
| - GBLM | 24.60 | 44.29 | 38.37 | 70.51 | 61.80 | 62.84 | 31.40 | 24.00 | 26.87 G |
| - PRUNER-ZERO | 24.02 | 44.77 | 37.42 | 71.22 | 62.75 | 62.33 | 32.76 | 24.00 | 26.87 G |
| - **MaskPro** | 21.10 | 46.81 | 38.76 | 71.60 | 64.25 | 64.23 | 33.19 | 24.80 | 35.90 G |
| **LLAMA-2-7B** | 8.71 | 57.15 | 39.62 | 78.07 | 68.90 | 76.35 | 43.34 | 31.40 | — |
| - MASKLLM | 12.55 | 51.17 | 38.56 | 74.70 | 65.04 | 69.57 | 35.67 | 26.80 | 331.16 G |
| - MAGNITUDE | 307.39 | 45.43 | 31.48 | 70.08 | 60.93 | 61.87 | 30.20 | 21.80 | 12.82 G |
| - SPARSEGPT | 21.07 | 43.20 | 36.56 | 70.89 | 64.56 | 64.52 | 31.48 | 24.60 | 22.20 G |
| - WANDA | 23.44 | 41.32 | 35.89 | 70.46 | 62.12 | 62.79 | 30.20 | 24.20 | 21.25 G |
| - GBLM | 21.64 | 41.79 | 34.61 | 70.57 | 62.75 | 63.17 | 29.86 | 23.20 | 26.87 G |
| - PRUNER-ZERO | 22.09 | 41.17 | 34.64 | 70.18 | 62.35 | 61.32 | 27.05 | 22.80 | 26.87 G |
| - **MaskPro** | 17.17 | 46.18 | 37.13 | 73.07 | 65.82 | 66.12 | 32.85 | 26.20 | 35.90 G |
| **DEEPSEEK-7B** | 9.70 | 56.94 | 39.62 | 79.27 | 70.40 | 75.25 | 43.60 | 32.60 | — |
| - MASKLLM | 12.90 | 51.73 | 39.14 | 75.95 | 65.80 | 68.10 | 35.32 | 25.80 | 339.56 G |
| - MAGNITUDE | 285.06 | 40.97 | 28.52 | 69.75 | 60.06 | 54.92 | 27.56 | 20.80 | 13.13 G |
| - SPARSEGPT | 19.12 | 45.58 | 37.80 | 73.94 | 65.43 | 66.37 | 32.94 | 24.80 | 22.50 G |
| - WANDA | 19.68 | 45.38 | 35.12 | 73.56 | 63.14 | 65.49 | 32.00 | 22.80 | 21.55 G |
| - GBLM | 19.55 | 45.34 | 36.17 | 73.99 | 62.98 | 65.82 | 32.85 | 23.60 | 27.98 G |
| - PRUNER-ZERO | 20.71 | 44.93 | 35.22 | 73.23 | 62.12 | 64.94 | 30.89 | 23.20 | 27.98 G |
| - **MaskPro** | 17.97 | 47.78 | 37.75 | 74.72 | 65.59 | 66.74 | 33.49 | 28.60 | 36.82 G |

---

**Theorem 2** *The proposed PGEs are all unbiased estimators of the policy gradient, i.e.,*

$$\mathbb{E}\left[g_p\right] = \mathbb{E}\left[g_r\right] = \mathbb{E}\left[g_{sr}\right] = \nabla\Phi(\pi). \tag{13}$$

*Furthermore, when the sampled mask satisfies $f(\mathbf{m}_t \odot \mathbf{w}, \xi) > \frac{1}{2}f(\mathbf{m}_0 \odot \mathbf{w}, \xi)$, we have:*

$$Var\left[g_{sr}\right] \lesssim Var\left[g_r\right] < Var\left[g_p\right]. \tag{14}$$

---

In Theorem 2, Eq.13 showes that our proposed updates $g_r$ and $g_{sr}$ are both unbiased estimators of the gradient $\nabla\Phi(\pi)$, effectively supporting the training process. Furthermore, when $f(\mathbf{m}_t \odot \mathbf{w}, \xi) > \frac{1}{2}f(\mathbf{m}_0 \odot \mathbf{w}, \xi)$, from Eq.14 of Theorem 2, we know that before the loss of the sampling mask $\mathbf{m}_t$ decreases to less than half of the initial one, using the update via loss residual with smoothing tracker can achieve more efficient training. Once the optimization process has sufficiently progressed such that the loss is less than half of the initial loss, a new set of $\mathbf{m}$ can be selected to replace $\mathbf{m}_0$ to continue efficient training. In practical experiments, this condition is almost easily satisfied, as the loss rarely drops below half of the initial value when training with an initial mask with simple priors.

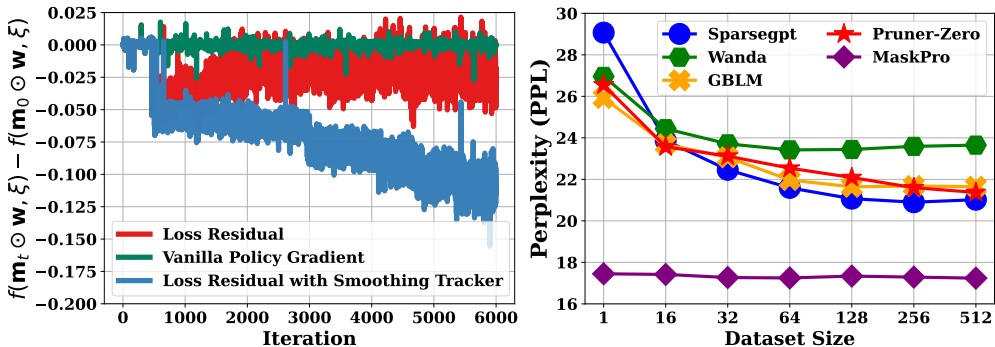

(a) Training Effectiveness of Three PGE Updates.  (b) Training Performance of Different Dataset Size.

Figure 3: (a) We show the different loss curves trained with the three PGEs. (b) We report the PPL on Wikitext of different methods trained with 1, 16, 32, 64, 128, 256, and 512 data samples.

## 6 EXPERIMENTS

In this section, we first introduce the baselines along with details of the dataset and models. Then we present the main experiments. We also conduct sensitivity studies of $\alpha$ and $C$ on Appendix A.9 and A.10 to provide proper guidance for the reproducibility and extensibility.

**Baselines.** We select the backpropagation-free methods including Magnitude (Han et al., 2015), SparseGPT (Frantar & Alistarh, 2023), Wanda (Sun et al., 2023), GBLM-Pruner (Das et al., 2023), and Pruner-Zero (Dong et al., 2024) as baselines. We also report the results of the backpropagation-based MaskLLM (Fang et al., 2024). The backpropagation-free methods perform sparsification by minimizing the layer-wise errors of the output activations caused by sparse weights, while MaskLLM updates the mask by optimizing masks through the loss function of the text generation task.

**Models & Dataset.** We evaluate the performance on 4 LLMs, including Vicuna-7B (Chiang et al., 2023), LLaMA-2-7B (Touvron et al., 2023), Deepseek-7B (DeepSeek-AI, 2024), Gemma-7B (Team et al., 2024). To ensure a fair comparison, we use the C4 dataset (Raffel et al., 2020) as a unified calibration or training dataset for each method and adopt the *LM-evaluation-harness* framework (Gao et al., 2024a) for zero-shot evaluations. Due to the page limitation, more details of the hyperparameters and experimental setups for reproducibility can be found in Appendix A.1.

**Performance.** In Table 1, we report the zero-shot evaluation on several downstream tasks for the (2:4)-sparsity. We conduct extensive experiments on several 7B models to validate the effectiveness of our proposed method. MaskPro generally outperforms existing non-backpropagation methods, achieving an average performance improvement of over 2% over the top-2 accuracy. On certain models and datasets, it achieves performance nearly comparable to MaskLLM. On the Wikitext PPL test, the MaskPro method also shows a consistent improvement, about 3 on LLaMA-2-7B and over 3 on the others. The weights of the Gemma-7B model are not sufficiently sparse, resulting in suboptimal performance of its corresponding sparse model and unstable PPL results. We show more evaluations in the Appendix A.3. More experiments of (4:8) / (8:16)-Sparsity are stated in Appendix A.5 and A.7. We also evaluate MaskPro on 13B and 30B models in Appendix A.8.

**Optimizers.** In Figure 3 (a), we evaluate the training performance of vanilla PGE, loss residual and loss residual with the smoothing tracker. The metric on the y-axis represents how much the loss value of the current minibatch is reduced by the mask sampled from the current logits compared to the initial mask. It can be observed that the vanilla policy gradient update is almost ineffective, with the loss oscillating around zero without effectively learning any useful information. After applying the loss residual update, significant improvement is observed as the logits receive effective guidance to sample better masks. However, its effect is not sufficiently stable — after achieving a certain level of improvement, large oscillations occur, preventing further learning progress. The update of loss residual with the smoothing tracker can efficiently and stably train this task, leading to better results.

**Size of Training Set.** Our proposed MaskPro requires significantly less data samples compared to other learning-based methods. As shown in Figure 3 (b), we evaluate the PPL of the Wikitext dataset on LLaMA2-7B after training 10k iterations with training set sizes of 1, 16, 32, 64, 128, 256, 512.

According to the experimental results reported by Fang et al. (2024), MaskLLM requires at least 1280 training samples to achieve the results of SparseGPT, and 520k samples for convergence. In contrast, our proposed MaskPro can be trained with a minimal number of training samples while maintaining nearly stable performance even with 1 data sample. We also provide results in Appendix A.12 comparing runs initialized from different masks with 1 sample versus 128 samples. Our experiments show that training with a single sample remains stable, with only a slight loss in performance.

**Training Efficiency.** We evaluate efficiency primarily by comparing memory usage, training time, and the size of the training dataset. Traditional rule-based methods learn masks by evaluating specific metrics on a small validation set. For example, in the (2:4)-sparsity on LLaMA-2-7B, the Pruner-Zero requires 26.87 GB of memory and 128 C4-en data samples. And for the learning-based MaskLLM, it requires 330 GB of memory across $8\times$ A100 GPUs and 520k training samples, taking over 1200 GPU hours. A significant advantage of our proposed MaskPro method is its low computational and memory overhead during training. More details of training time profile reports on different patterns and corresponding acceleration techniques of sampling are shown in Appendix A.13 and A.14.

## 7 SUMMARY

In this paper, we propose a novel memory-efficient framework named MaskPro, which leverages policy gradient updates to learn semi-structured sparsity. By reformulating the (N:M)-sparsity as a linear-space probability relaxation, our approach reduces the memory for logits storage from vanilla $\mathcal{O}\left(\binom{M}{N}\frac{d}{M}\right)$ to $\mathcal{O}(d)$. Furthermore, we propose a novel PGE that replaces the vanilla loss metric with loss residuals, refined by a moving average tracker, effectively accelerating training and reducing variance. Lastly, comprehensive theoretical analysis and extensive experiments demonstrates the effectiveness of our MaskPro in achieving substantial performance gains with minimal training costs.

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

**The Use of Large Language Models.** In this work, we only evaluate the performance on LLMs in our experiments and employ LLMs to refine the writing and presentation of our manuscript. Other aspects of the work are unrelated to LLMs.

**Limitation and Broader Impact.** This paper presents a memory-efficient training framework for learning semi-structured sparse masks based on policy gradient, achieving comprehensive improvements in performance and efficiency through substantial upgrades in both the probabilistic modeling and optimizers. A limitation of this paper is that when training large-scale models, the primary time consumption lies in simulating the mask sampling process. Utilizing more efficient sampling simulations can further enhance training efficiency. The core contributions of this paper mainly include linear-space probabilistic modeling and optimizer enhancements. These two aspects can be widely applied to various model pruning tasks, not just the specific task addressed in this work.

# A EXPERIMENTS

## A.1 EXPERIMENTAL DETAILS AND REPRODUCIBILITY

In this paper, we reproduce the baselines using their official open-source codes provided in each paper. For fairness, we use the C4-en dataset as the calibration/training dataset. For the MaskLLM, we follow Fang et al. (2024) to adopt 520k C4-en samples for training 2k iterations with batchsize 256. For other methods, we follow their setups to adopt 128 C4-en samples as calibration dataset.

**Hyperparameters.** For the MaskPro, we evaluate a wide range of dataset sizes, ranging from 1 to 320k. We select the learning rate from $[25, 50, 100, 200]$ for each model and $50/100$ proves to be a relatively effective choice. In the training, we use batchsize as 32 and training for $\sim$10k iterations. Using a batchsize larger than 32 is also encouraged, as larger batches generally lead to stable training. In all experiments, we adopt the smoothing coefficient $\alpha = 0.99$ to stably follow the loss residual. We summarize the selection of certain hyperparameters in Table 2.

Table 2: Hyperparameters selections.

| Model | Learning rate | Logits Magnitude | Smoothing coefficient $\alpha$ | Initial Mask |
|---|---|---|---|---|
| Gemma-7B | 50 / 100 | 10.0 | 0.99 | Top-$N$ / Sparsegpt |
| Vicuna-V1.3-7B | 50 | 10.0 | 0.99 | Top-$N$ / Sparsegpt |
| LLaMA-2-7B | 50 | 10.0 | 0.99 | Top-$N$ / Sparsegpt |
| DeepSeek-7B | 50 / 100 | 10.0 | 0.99 | Top-$N$ / Sparsegpt |

**Initialization.** The initialization of logits in MaskPro is crucial. **Standard random initialization or zero initialization are ineffective.** This is because the logits determine the sampling scale. For instance, zero initialization implies that each position is sampled with equal probability, leading to a very large number of negative samples during the initial training stage. Consequently, it becomes exceedingly difficult to identify effective positive samples for learning. In our experiments, we initialize the logits based on $\pi_0 = \mathbf{m}_0 * C$, where $\mathbf{m}_0$ is a pre-defined mask and $C$ is the initial logits magnitude. A larger $C$ indicates that the mask changes less compared to the initial mask $\mathbf{m}_0$, effectively maintaining a balance between positive and negative samples in the early training stages. The design of $\mathbf{m}_0$ is flexible. In practice, training can also start with a randomly generated mask; however, this approach typically requires a longer training period. We recommend directly using the results from the Sparsegpt method or selecting the Top-$N$ positions over $M$ elements per group.

**Training Environment.** We train our proposed MaskPro on a single H100 / A100 GPU device. Other details are stated in Table 3.

Table 3: Training Environment.

| GPU | CPU | CUDA | Driver | Pytorch |
|---|---|---|---|---|
| 1× H100 / A100 | 128× AMD EPYC 9354 32-C | 12.4 | 535.230.02 | 2.5.1 |

**Evaluations.** For fair comparisons, all evaluations are conducted on the public benchmark framework *LM-evaluation-harness framework* (Gao et al., 2024a) (https://github.com/EleutherAI/lm-evaluation-harness.git). Please refer to the relevant reproduction guidelines.

## A.2 THE IMPORTANCE OF $C$ IN LOGITS INITIALIZATION

We have previously discussed the selection of $C$ in the experiments. Here, we will visualize some practical scenarios encountered during the experiments and illustrate why $C$ must be sufficiently large to effectively drive the training process. We analyze the distribution of loss values of training LLAMA-2-7B within 100 steps with a minibatch of 32 samples under different $C$ initialization settings, as shown in Figure 4.

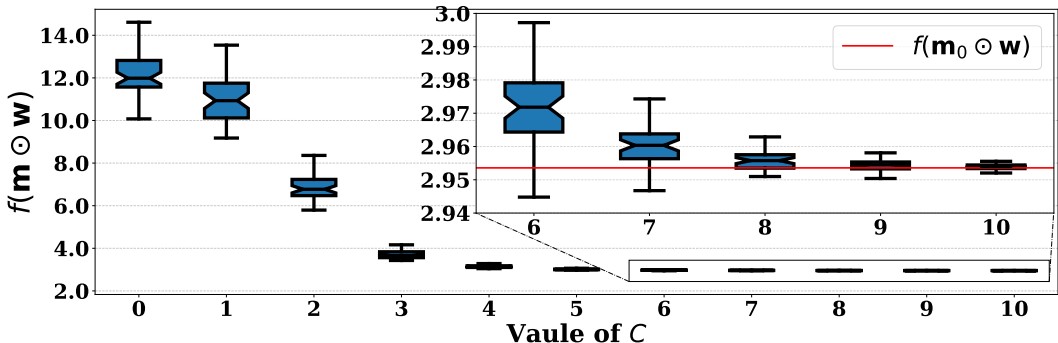

Figure 4: The distribution of loss within 100 steps under different $C$ used for logits initialization.

**We first explain which variables are affected by $C$.** Since we use the softmax function to generate the probabilities for the corresponding positions, the logits values determine whether the initial probability of being sampled at a specific position is sufficiently large. In other words, when sampling a new mask, it ensures how many positions with high probabilities remain unchanged. This point is particularly important because the sampling space is extremely large. Without constraining the sampling space, there is a high probability of sampling poor masks. Extremely poor masks are incapable of capturing useful information effectively. Therefore, randomly initializing the $C$ value or directly setting it to zero is completely ineffective, as it cannot ensure the stability of the sampling space, i.e., whether the distribution of positive and negative samples in the sampling space is balanced.

**Next, we explain the meaning of Figure 4.** We show the distribution of loss values over 100 training steps using a minibatch under different $C$ initialization settings on the LLaMA-2-7B model. In the subplot, the red line corresponds to the loss of the initialized mask $\mathbf{m}_0$. When $C$ is small, it is evident that the training fails — the loss surges from the initial 2.95 to over 10. A large number of negative samples flood into the training process, leading to chaotic learning. As $C$ increases to 4, the stability gradually improves. However, it is still insufficient. As shown in the subplot, even when $C = 6$, more than 90% of the sampled masks still exhibit extremely poor performance. Until $C$ increases to 9 and 10, it can be observed that the distribution of positive and negative sampled masks during training gradually maintains a 1:1 ratio. By this, the training can proceed effectively.

Here, we provide an additional example to explain and guide the selection of $C$ for different network parameters. As mentioned earlier, one probabilistic interpretation of $C$ is to determine, on average, how many positions are sampled differently from the initialized mask. We can succinctly express this probability in a mathematical form. Suppose the initialized mask $\mathbf{m}_0$ is $[0, 1, 1, 0]$, then its initial logits is $[0, C, C, 0]$ and the corresponding softmax probability is $\left[\frac{1}{2(e^C+1)}, \frac{e^C}{2(e^C+1)}, \frac{e^C}{2(e^C+1)}, \frac{1}{2(e^C+1)}\right]$. Thus we have:

$$p(\mathbf{m} = [0, 1, 1, 0] \,|\, \pi = [0, C, C, 0]) = \frac{e^{2C}}{(e^C + 1)(e^C + 2)}.$$

In fact, the size of the sampling space where positive and negative samples are evenly distributed is difficult to estimate for different model parameter sizes. However, we can reasonably speculate that

the total number of parameters is generally proportional to the above probability value. For larger models, using a larger $C$ can further maintain the effectiveness of the training space.

### A.3 MORE EXPERIMENTS ON DIFFERENT TASKS

In addition to the primary comparisons presented in the main text, we extend our evaluation to encompass over a dozen additional tasks to provide a more comprehensive demonstration of the effectiveness of our proposed method. These extended tests are carefully selected to cover diverse data distributions and task complexities, allowing us to assess the robustness and generalizability of our approach. The results from these comprehensive experiments consistently highlight the superior performance of our method across various scenarios, further reinforcing its effectiveness. The detailed outcomes of these evaluations are presented as follows.

Table 4: Zero-shot evaluations of (2:4)-sparsity on other more tasks.

| | LLaMA-2-7B | | | | DeepSeek-7B | | | |
|---|---|---|---|---|---|---|---|---|
| | **Dense** | **Sparsegpt** | **Pruner-Z** | **MaskPro** | **Dense** | **Sparsegpt** | **Pruner-Z** | **MaskPro** |
| **WMDP** | 39.29 | 26.61 | 26.52 | **26.95** | 41.00 | 27.15 | 27.07 | **28.22** |
| **TMLU** | 29.58 | 25.03 | 25.13 | **25.38** | 37.17 | **25.99** | 24.36 | 25.37 |
| **Prost** | 23.60 | 24.26 | 24.03 | **24.41** | 28.19 | 28.22 | 27.62 | **29.57** |
| **AExams** | 21.04 | **23.65** | **23.65** | **23.65** | 23.65 | **23.65** | **23.65** | **23.65** |
| **AClue** | 27.47 | 25.33 | 25.31 | **26.24** | 32.34 | 27.17 | 26.88 | **27.31** |
| **ANLI-1** | 36.40 | 33.20 | 33.60 | **34.40** | 34.10 | 31.10 | 31.19 | **32.20** |
| **ANLI-2** | 37.20 | 34.10 | 33.90 | **34.10** | 36.60 | **33.70** | 33.20 | 33.50 |
| **ANLI-3** | 37.58 | 33.08 | 33.00 | **35.67** | 37.75 | 33.33 | 33.04 | **33.85** |
| **SCIQ** | 94.00 | **91.10** | **91.10** | **91.10** | 94.10 | **92.30** | 90.20 | 90.90 |
| **MathQA** | 28.24 | 23.72 | 23.55 | **23.95** | 29.48 | 25.93 | 25.12 | **26.76** |
| **Haerae** | 22.27 | 18.88 | **18.91** | 18.79 | 29.70 | **25.57** | 18.26 | 22.18 |
| **BoolQ** | 77.68 | 71.10 | 69.13 | **71.12** | 72.81 | 66.91 | 66.36 | **67.77** |
| **ComQA** | 32.92 | 20.80 | 20.08 | **22.03** | 36.69 | 23.10 | 22.95 | **23.18** |
| **LogiQA** | 25.65 | 21.66 | 21.78 | **22.89** | 25.04 | 21.73 | 21.35 | **22.58** |
| **COPA** | 87.00 | **81.00** | 79.00 | 79.00 | 84.00 | 86.00 | 84.00 | **87.00** |
| **WIC** | 49.84 | 47.81 | 47.22 | **49.84** | 51.10 | 48.00 | 48.81 | **49.06** |
| **WSC** | 36.54 | **36.54** | **36.54** | **36.54** | 64.42 | **36.54** | **36.54** | **36.54** |
| **CB** | 42.86 | 41.07 | 39.29 | **57.14** | 55.36 | 42.86 | 43.44 | **48.21** |
| **MultiRC** | 56.97 | **57.20** | 56.37 | 56.93 | 57.22 | **57.20** | **57.20** | **57.20** |
| **RTE** | 62.82 | 58.48 | 59.12 | **61.37** | 67.87 | 63.43 | 63.15 | **66.32** |
| **Mutual** | 70.84 | 68.01 | 67.44 | **68.53** | 71.30 | 67.43 | 67.24 | **68.33** |
| **WebQS** | 0.0586 | 0.0541 | 0.0544 | **0.0566** | 0.0876 | 0.0468 | 0.0226 | **0.0494** |

In this experiment, we evaluate the performance of MaskPro across a diverse set of tasks to comprehensively assess its effectiveness on LLaMA-2-7B and DeepSeek-7B. The experimental design includes a variety of downstream tasks. MaskPro consistently demonstrates superior performance over competing methods, such as SparseGPT and Pruner-Zero, in the majority of datasets. The method effectively balances accuracy and computational efficiency, achieving more favorable outcomes without compromising on memory constraints. This consistent performance across multiple tasks highlights the robustness and generalizability of MaskPro in handling different scenarios. On smaller datasets, the performance gains of MaskPro are relatively moderate, as the evaluation is constrained by limited sample diversity. However, when tested on larger datasets with extensive testing samples, MaskPro consistently demonstrates substantial improvements over baseline methods.

## A.4 Training with Different Dataset Size

In this section, we report the training results using different numbers of samples. In Figure 5, we present the loss residuals of training on LLaMA-2-7B model with 1, 32, 128, and 320k samples, respectively. We set batchsize as 32 for all others expect for 1 as 1. All are trained for 10k iterations.

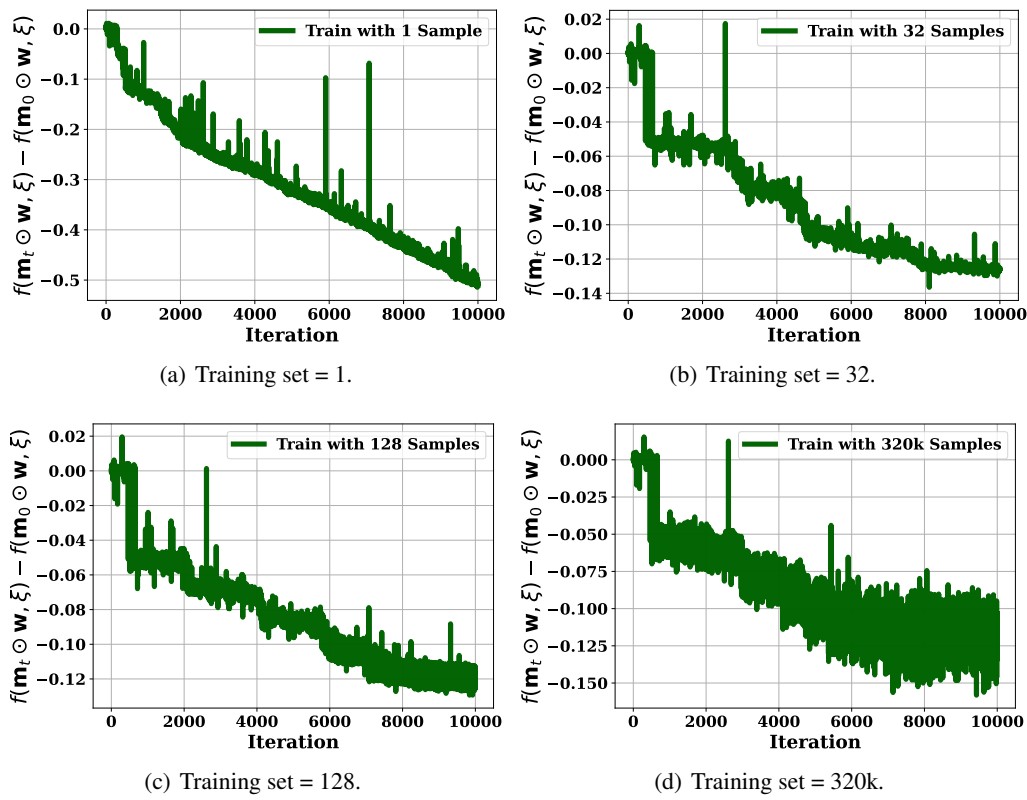

(a) Training set = 1.  (b) Training set = 32.

(c) Training set = 128.  (d) Training set = 320k.

Figure 5: Loss residual curves of training on LLaMA-2-7B model with 1, 32, 128, and 320k samples.

It can be observed that MaskPro does not require a large number of training samples. Even with just 1 sample (in a single minibatch), it can complete training and achieve stable performance. The loss on a single training sample can steadily decrease, but this does not necessarily imply a continually decreased loss on the test dataset. In fact, despite the persistent reduction in training loss, the test set performance may have already stabilized. In Figure 3 (b) of the main text, we report the testing results of the learned mask on the Wikitext dataset. Next, we evaluate the zero-shot accuracy on a series of downstream tasks, shown in Table 5.

Table 5: Zero-shot evaluations of masks trained with different dataset size on LLaMA-2-7B.

|  | HellaS. | RACE | PIQA | WinoG. | ARC-E | ARC-C | OBQA | Avg. |
|---|---|---|---|---|---|---|---|---|
| 320k samples | 46.18 | 37.13 | 73.07 | 65.82 | 66.12 | 32.85 | 26.20 | 49.62 |
| 128 samples | 46.10 | 37.03 | 72.47 | 65.62 | 65.49 | 32.25 | 25.80 | 49.25 |
| 32 samples | 46.32 | 36.89 | 72.80 | 65.27 | 65.95 | 32.66 | 25.80 | 49.38 |
| 1 sample | 46.39 | 37.61 | 72.96 | 64.64 | 65.70 | 32.59 | 24.40 | 49.18 |

It can be observed that although the performance slightly declines, overall, even training with just 1 sample can still maintain satisfactory results, and in some datasets, the performance is even slightly higher.

## A.5 PERFORMANCE OF (4:8)-SPARSITY

In this section, we report the results for (4:8)-sparsity in Table 6 and corresponding training loss curves in Figure 6. The training hyperparameters are consistent with those reported in Table 2.

Table 6: Zero-shot evaluations of (4:8)-sparsity. The MaskLLM method suffers from severe memory explosion and exceeds the memory limitation of $8\times$ A100 GPUs ($> 640$ G).

|  | Wiki. | HellaS. | RACE | PIQA | WinoG. | ARC-E | ARC-C | OBQA |
|---|---|---|---|---|---|---|---|---|
| **LLAMA-2-7B** | 8.71 | 57.15 | 39.62 | 78.07 | 68.90 | 76.35 | 43.34 | 31.40 |
| - MASKLLM | — | — | — | — | — | — | — | — |
| - MAGNITUDE | 61.99 | 46.05 | 35.31 | 72.20 | 62.27 | 64.81 | 34.07 | 25.80 |
| - SPARSEGPT | 14.99 | 48.19 | 38.55 | 73.78 | 67.72 | 68.15 | **36.01** | 27.80 |
| - WANDA | 15.28 | 47.04 | 38.18 | 74.14 | 66.77 | 67.00 | 34.56 | 26.40 |
| - GBLM | 15.21 | 47.32 | 37.51 | 74.16 | 67.56 | 67.13 | 34.56 | 27.20 |
| - PRUNER-ZERO | 15.10 | 47.82 | 38.13 | 74.07 | 67.23 | 68.18 | 34.97 | 27.20 |
| - **MaskPro** | **13.73** | **49.51** | **39.33** | **74.65** | **68.43** | **68.64** | 35.92 | **28.20** |
| **DEEPSEEK-7B** | 9.70 | 56.94 | 39.62 | 79.27 | 70.40 | 75.25 | 43.60 | 32.60 |
| - MASKLLM | — | — | — | — | — | — | — | — |
| - MAGNITUDE | 109.37 | 45.32 | 32.06 | 72.42 | 61.64 | 56.31 | 32.68 | 23.40 |
| - SPARSEGPT | 14.67 | 48.36 | 38.09 | 75.24 | 65.82 | **70.20** | 36.69 | 29.20 |
| - WANDA | 14.76 | 49.09 | 38.47 | 75.46 | 64.88 | 68.48 | 34.22 | 27.20 |
| - GBLM | 14.74 | 49.03 | 38.76 | 75.73 | 65.11 | 68.18 | 34.13 | 27.00 |
| - PRUNER-ZERO | 14.85 | 48.22 | 38.32 | 75.12 | 65.66 | 69.23 | 35.50 | 27.80 |
| - **MaskPro** | **13.89** | **50.97** | **39.25** | **75.87** | **66.27** | 69.51 | **36.89** | **29.80** |

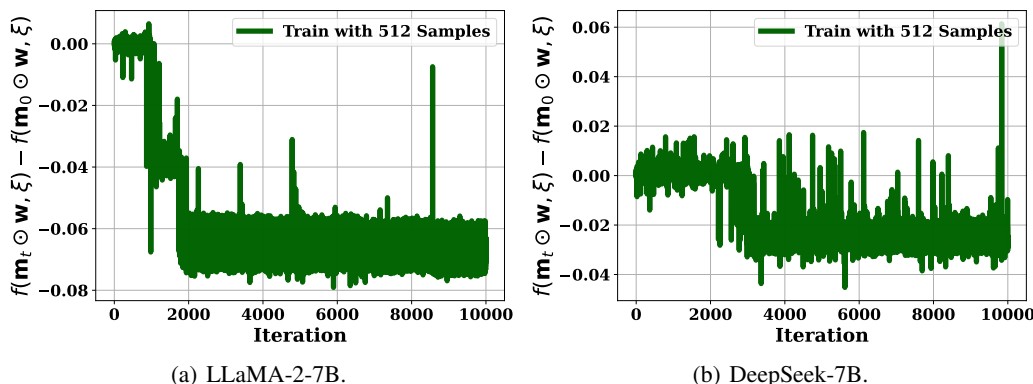

(a) LLaMA-2-7B.           (b) DeepSeek-7B.

Figure 6: Loss residual curves of training for the (4:8)-sparsity.

## A.6 MEMORY SCALABILITY

In this section, we report the memory scalability in Table 7.

Table 7: Memory (GB) reuqired for training on DeepSeek-7B.

|  | MaskLLM | MaskPro |
|---|---|---|
| (1:4)-Sparsity | 266.35 | 36.82 |
| (2:4)-Sparsity | 339.56 | 36.82 |
| (4:8)-Sparsity | >640.00 | 36.95 |

The MaskPro method, due to its linear probability modeling, almost does not cause memory growth as the (N:M) ratio scales. When training the (4:8)-sparsity on DeepSeek-7B model, MaskLLM has

encountered OOM (Out of Memory) on $8\times$ A100 ($>$640G). In contrast, MaskPro can achieve the expansion with almost no additional memory overhead.

## A.7 PERFORMANCE OF (8:16)-SPARSITY

Moreover, we provide the (8:16)-Sparsity pattern to evaluate the performance of our proposed MaskPro method. This setting involves significantly larger combinatorial spaces which can greatly support the efficiency of MaskPro.

Table 8: Zero-shot evaluations of (8:16)-sparsity on LLaMA2-7B.

|  | HellaS. | RACE | PIQA | WinoG. | ARC-E | ARC-C | OBQA | Avg. |
|---|---|---|---|---|---|---|---|---|
| **LLAMA-7B** | 57.15 | 39.62 | 78.07 | 68.90 | 76.35 | 43.34 | 31.40 | 56.40 |
| - MAGNITUDE | 52.27 | 35.02 | 72.74 | 64.48 | 67.68 | 37.03 | 27.20 | 50.92 |
| - SPARSEGPT | 50.19 | 39.04 | 74.43 | 66.22 | 70.45 | 36.43 | 28.80 | 52.22 |
| - WANDA | 49.77 | 39.14 | 75.30 | **66.61** | 70.62 | 36.18 | 28.80 | 52.35 |
| - GBLM | 49.51 | **39.90** | 75.68 | 66.38 | 69.91 | 36.43 | 27.60 | 52.20 |
| - PRUNER-ZERO | 50.12 | 38.68 | 75.22 | 66.13 | 69.93 | 35.48 | 27.80 | 51.91 |
| - **MaskPro** | **53.15** | 39.23 | **76.15** | 66.56 | **72.87** | **40.13** | **29.60** | **53.96** |
| **LLAMA-13B** | 60.05 | 40.48 | 79.11 | 72.22 | 79.42 | 48.46 | 35.20 | 59.28 |
| - MAGNITUDE | 55.43 | 37.51 | 74.48 | 66.06 | 68.94 | 38.05 | 27.60 | 52.58 |
| - SPARSEGPT | 54.24 | **40.38** | 77.15 | 70.19 | 75.08 | 41.31 | **31.00** | 55.62 |
| - WANDA | 54.50 | 39.62 | 77.09 | 70.09 | 73.19 | 40.36 | 30.80 | 55.09 |
| - GBLM | 54.45 | 39.18 | 76.35 | 69.92 | 73.75 | 40.07 | 29.60 | 54.76 |
| - PRUNER-ZERO | 54.11 | 38.64 | 76.28 | 70.41 | 72.92 | 40.55 | 30.00 | 54.70 |
| - **MaskPro** | **57.35** | 39.92 | **77.83** | 70.68 | **76.45** | **43.26** | 30.60 | **56.58** |

Under this sparsity pattern, the memory requirement of MaskLLM becomes extremely large, even exceeding the resource demands commonly used in the community to train models with hundreds of billions of parameters. Moreover, our MaskPro approach introduce minor training cost, while achieving better results than rule-based methods.

## A.8 PERFORMANCE ON LARGER SCALE MODELS

In this section, we present the results of applying MaskPro to larger models, specifically the 13B and 30B variants. We retain the same hyperparameter settings used for the 7B model, with the only adjustment being a slight tuning of the initialization logits magnitude.

Table 9: Zero-shot evaluations of (2:4)-sparsity on 13B/30B models.

|  | HellaS. | RACE | PIQA | WinoG. | ARC-E | ARC-C | OBQA | Avg. |
|---|---|---|---|---|---|---|---|---|
| **LLAMA-13B** | 60.05 | 40.48 | 79.11 | 72.22 | 79.42 | 48.46 | 35.20 | 59.28 |
| - MAGNITUDE | **50.10** | 36.84 | 71.76 | 61.88 | 62.29 | 31.74 | 23.40 | 48.29 |
| - SPARSEGPT | 47.73 | **38.95** | 73.61 | 69.22 | 69.95 | 36.35 | **27.40** | 51.89 |
| - WANDA | 46.24 | 38.47 | 73.94 | 67.32 | 68.73 | 34.13 | 24.20 | 50.43 |
| - GBLM | 46.65 | 37.97 | 73.46 | 69.04 | 69.33 | 34.75 | 25.80 | 51.00 |
| - PRUNER-ZERO | 46.15 | 38.85 | 73.13 | 67.24 | 67.52 | 33.89 | 25.20 | 50.28 |
| - **MaskPro** | 49.24 | 38.91 | **75.12** | **70.33** | **71.85** | **38.26** | **27.40** | **53.02** |
| **LLAMA-30B** | 63.36 | 39.14 | 80.63 | 75.85 | 80.64 | 51.45 | 36.40 | 61.07 |
| - MAGNITUDE | 49.57 | 35.69 | 70.24 | 65.59 | 57.32 | 31.66 | 27.80 | 48.27 |
| - SPARSEGPT | 55.25 | 37.77 | 77.45 | **73.68** | 75.25 | 43.27 | 31.80 | 56.35 |
| - WANDA | 54.18 | **40.00** | 77.69 | 73.24 | 74.24 | 42.15 | 31.60 | 56.16 |
| - GBLM | 54.68 | 37.35 | 75.24 | 73.12 | 74.68 | 42.32 | 30.80 | 55.46 |
| - PRUNER-ZERO | 53.69 | 37.13 | 75.86 | 73.04 | 74.23 | 41.25 | 31.20 | 55.20 |
| - **MaskPro** | **59.76** | 37.28 | **78.24** | 73.32 | **76.83** | **45.65** | **33.20** | **57.75** |

Notably, MaskPro remains highly effective even when applied to models at the 30B scale. This demonstrates the robustness and scalability of the proposed probabilistic formulation. Furthermore, due to the linear probability modeling and the use of policy-gradient–based optimization, MaskPro achieves this performance with significantly reduced computational overhead. In particular, the training process requires far fewer resources compared to methods that rely on dense mask representations or exhaustive combinatorial search. These properties highlight the practical advantages of MaskPro, especially in large-scale scenarios where both memory efficiency and training stability are critical.

## A.9  SENSITIVITY OF TRACKER COEFFICIENT $\alpha$

In this part, we demonstrate the sensitivity studies of the tracker coefficient $\alpha$. In our PG update, the parameter $\alpha$ is used to track a stable estimate of the current baseline and prevent it from being overly influenced by the stochastic variance of sampled losses. Conceptually, this plays the same role as $\beta_1$ or $\beta_2$ in the Adam optimizer. To examine its sensitivity, we conducted the following set of experiments:

Table 10: Sensitivity studies of tracker coefficient $\alpha$.

|  | $\alpha = 0.7$ | $\alpha = 0.9$ | $\alpha = 0.95$ | $\alpha = 0.99$ | $\alpha = 0.995$ |
|---|---|---|---|---|---|
| **LLAMA-7B** | 34.25 | 48.28 | 49.37 | **49.62** | 49.21 |
| **LLAMA-13B** | 38.68 | 51.23 | 52.78 | **53.02** | 52.74 |

We find that using $\alpha = 0.99$ consistently across all tasks provides the most stable and reliable performance. Therefore, we only report the selection of 0.99 for reproduction in the main text. This hyperparameter requires almost no additional tuning.

## A.10  SENSITIVITY OF LOGITS MAGNITUDE $C$

In this part, we demonstrate the sensitivity of the logits magnitude $C$. In the initialization, the parameter $C$ is used for stable sampling space. A detailed explanation is provided in Appendix A.2. If $C$ is set too small, a single sampling step has a high probability of producing a poor mask, which can lead to a severe imbalance between positive and negative samples during training, ultimately hindering the learning process of combinatorial optimization. Therefore, choosing a sufficiently large $C$ during initialization allows the training to remain stable. We evaluated different values and the results are as follows:

Table 11: Sensitivity studies of initial logits magnitude $C$.

|  | $C = 8$ | $C = 9$ | $C = 10$ | $C = 11$ | $C = 12$ |
|---|---|---|---|---|---|
| **LLAMA-7B** | - | 49.17 | **49.62** | 49.59 | 49.55 |
| **LLAMA-13B** | - | 52.45 | 52.94 | **53.02** | 52.99 |

## A.11  ABLATION STUDIES OF INITIAL MASK $m_0$

In this part, we evaluate how the different initialization mask $m_0$ affects the results. Unlike gradient-based methods, RL methods typically converge more slowly, so a good initialization can significantly shorten the training process.

Table 12: Ablation studies of initial mask $m_0$.

|  | Random | Top-K | Wanda | GBLM | Sparsegpt |
|---|---|---|---|---|---|
| **LLAMA-7B** | 30.27 | 45.97 | 46.71 | 46.56 | 47.97 |
| - MASKPRO | 36.35 | 48.35 | 49.33 | 49.45 | 49.62 |
| - IMPROVEMENT | +6.08 | +2.38 | +2.62 | +2.89 | +1.65 |

In practice, when using SparseGPT for initialization, the model converges in roughly 10000 steps. With TopK initialization, extending the training over 20000 steps yields a relatively smooth result. We can see that random initialization can also train the mask, but the training process is slow. We further conduct a longer training experiment specifically for random initialization, and the results are as follows:

Table 13: Long-term training on the random initialization.

|  | $T = 20000$ | $T = 30000$ | $T = 50000$ | $T = 70000$ |
|---|---|---|---|---|
| ACCURACY | 36.35 | 38.43 | 40.74 | 42.37 |

This training process is quite lengthy, and we estimate that completing the full experiment would require at least 300000 steps. Such behavior is consistent with the theoretical convergence rate of RL-based methods, which is why we do not encourage training from random initialization. We hope that these two experiments address the reviewer's concerns: it is not that RL-based methods cannot be trained from random initialization, but rather that it is unnecessary, as simple priors such as top-K can significantly shorten the training cycle.

## A.12 TRAINING WITH 1 DATA SAMPLE FROM DIFFERENT INITIAL MASK $m_0$

In this part, we additionally evaluate the stability of the training process of "with 1 data sample" from different initialization.

Table 14: Ablation studies of dataset size on different initial mask $m_0$.

|  | Random | Top-K | Wanda | GBLM | Sparsegpt |
|---|---|---|---|---|---|
| WITH 128 DATA SAMPLES | 36.35 | 49.35 | 49.33 | 49.45 | 49.62 |
| WITH 1 DATA SAMPLES | 35.97 | 49.12 | 49.04 | 49.21 | 49.18 |

We would like to clarify that MaskPro is indeed not very sensitive to the number of samples. The essence of RL-based methods lies in accurately estimating and constructing the reward, rather than relying on large data volumes. While we do not deny that using a larger dataset may yield further improvements, the performance obtained with only a few hundred samples is already very close.

## A.13 TRAINING TIME PROFILES ON MASKPRO

In this part, we mainly show the training time profiles of our MaskPro method under different patterns and models. We use torch.multinomial function for (N:M)-sparsity sampling, which simulates the sampling process through a lookup-based mechanism and provides high accuracy. The forward pass is implemented through a standard wrapper function. Specifically, we wrap the linear layer with an additional mask parameter and integrate the mask computation directly inside the linear operation. This design avoids modifying PyTorch's computation graph and enables efficient inference. The logits updates are computed entirely through matrix calculation, and PyTorch's built-in libraries already provide the necessary parallelization. To further illustrate the implementation details, we report the per-step training time as follows:

Table 15: Averaged time required in each step on (2:4)-Sparsity.

|  | Mask Sampling | | Forward | | PG Update | |
|---|---|---|---|---|---|---|
|  | TIME | RATIO | TIME | RATIO | TIME | RATIO |
| **LLAMA-7B** | 2.328s | 85.94% | 0.062s | 2.29% | 0.319s | 11.77% |
| **LLAMA-13B** | 4.739s | 85.83% | 0.139s | 2.52% | 0.644s | 11.65% |

Similar to the general RL method (Fan et al., 2024), the main source of time consumption comes from the sampling process. We also evaluated the sampling performance under different sparsity patterns, as shown in the table below.

Table 16: Mask sampling time in each step on different (N:M)-Sparsity.

|  | (2:4)-Sparsity | (4:8)-Sparsity | (8:16)-Sparsity |
|---|---|---|---|
| **LLaMA-7B** | 2.328s | 1.334s | 0.794s |
| **LLaMA-13B** | 4.739s | 2.692s | 1.574s |

We can observe that doubling the model size roughly doubles the sampling time. Another interesting observation is that the sampling time of (N:M)-Sparsity depends on $M$. With the same model size, a larger $M$ leads to shorter sampling time, due to parallel optimizations in the sampling process. For a $d$-dimensional model, there are a total of $\frac{d}{M}$ sampling groups. Although increasing $N$ and $M$ makes each group more expensive to sample, the total number of parallel groups decreases proportionally. This reduction in the number of groups results in a more favorable computation pattern for hardware. Consequently, for more complex (N:M)-Sparsity, the time required for a single sampling step can actually be lower. Large M is a GPU-friendly selection.

### A.14 Alternative Strategies for Accelerating Sampling

In this part, we additionally explore two alternative accelerated sampling strategies along with their corresponding results. Sampling is the primary computational bottleneck of RL-based methods. Therefore, we explored several alternative acceleration strategies to speed up the training process, and their effects are summarized below.

Table 17: Acceleration of mask sampling and their corresponding performance on (2:4)-Sparsity.

|  | torch.multinomial | | Naive Gumbel-TopK | | Gaussion-TopK | |
|---|---|---|---|---|---|---|
|  | TIME | ACC. | TIME | ACC. | TIME | ACC. |
| **LLaMA-7B** | 2.328s | 49.62 | 1.821s (1.27×) | 49.24 (-0.38) | 1.496s (1.58×) | 48.84 (-0.78) |
| **LLaMA-13B** | 4.739s | 53.02 | 3.645s (1.30×) | 52.59 (-0.43) | 3.061s (1.55×) | 52.22 (-0.80) |

Within an acceptable error range, the training time can be further reduced. However, we still recommend using higher-precision sampling methods, as the current training time requirement of MaskPro is already quite reasonable. On the impact of randomness on experiments, RL methods rely on sampling, so they are generally less sensitive to random seeds compared with gradient-based methods, and tend to exhibit stronger robustness across settings.

## B Detailed Description of Without-Replacement Probability $p(\mathbf{m}|\pi)$

This section aims to present a specific form of $p(\mathbf{m}|\pi)$ and its related gradient $\nabla \log(p(\mathbf{m}|\pi))$. Note that in Eq.8, we define $p(\mathbf{m}|\pi) := \prod_{i=1}^{\frac{d}{M}} p(\mathbf{m}_i|\pi_i)$ where $\mathbf{m}_i := \bigoplus_{j=1}^{N} \mathbf{a}_{i,j}$ for any $i \in [\frac{d}{M}]$ and $p(\mathbf{m}_i|\pi_i)$ denotes the probability of our MaskPro generating the mask $\mathbf{m}_i$ under logits $\pi_i$. Therefore, before presenting the details of $p(\mathbf{m}|\pi)$, we firstly investigate the probability $p(\mathbf{m}_i|\pi_i)$.

### B.1 Detailed Description of $p(\mathbf{m}_i|\pi_i)$

It is worth noting that the mask vector $\mathbf{m}_i \in \mathcal{S}^{N:M}$ such that we can assume $\mathbf{m}_i = \sum_{i \in [N]} \mathbf{e}_{k_i}$ where $\mathbf{e}_j$ denotes the $j$-th basis vector of the space $\mathbb{R}^{1 \times M}$, $k_i \in [M], \forall i \in [N]$ and $k_1 \neq k_2 \neq \ldots \neq k_N$. In other words, $\{k_1, \ldots, k_N\}$ is an $N$-size subset of $[M] = \{1, \ldots, M\}$.

From the definition of $\mathbf{m}_i$, we know that $\mathbf{m}_i := \bigoplus_{j=1}^{N} \mathbf{a}_{i,j}$. Furthermore, according to Eq.23 in Section C.1, we also can know that, in order to ensure that $\mathbf{m}_i = \bigoplus_{j=1}^{N} \mathbf{a}_{i,j} = \sum_{i \in [N]} \mathbf{e}_{k_i}$, we typically require a one-to-one assignment of the previously defined $N$ distinct basis vectors $\{\mathbf{e}_{k_1}, \ldots, \mathbf{e}_{k_N}\}$ to $\{\mathbf{a}_{i,1}, \ldots, \mathbf{a}_{i,N}\}$. In general, there are $N!$ different ways to perform this matching.

To better illustrate our results, we introduce the concept of permutation from group theory to represent these $N!$ one-to-one assignment. More specifically, for any one-to-one assignment from $\{\mathbf{e}_{k_1}, \ldots, \mathbf{e}_{k_N}\}$ to $\{\mathbf{a}_{i,1}, \ldots, \mathbf{a}_{i,N}\}$, we represent is as a bijective function $\sigma : \{1, \ldots, N\} \rightarrow \{k_1, \ldots, k_N\}$. Here, each bijection $\sigma$ means that we match each basis vector $\mathbf{e}_{\sigma(j)}, \forall j \in [N]$ to the $j$-th sampled vector $\mathbf{a}_{i,j}$ in the sampling-without-replacement process, namely, $\mathbf{a}_{i,j} = \mathbf{e}_{\sigma(j)}, \forall j \in [N]$. Moreover, we denote all such bijections as $B_N(\mathbf{m}_i)$, that is to say,

$$B_N(\mathbf{m}_i) := \{\sigma : \sigma \text{ is a bijection from } [N] \text{ to } \{k_1, \ldots, k_N\}\}.$$

With the notions of $\sigma$ and $B_N(\mathbf{m}_i)$, we next present the specific form of $p(\mathbf{m}_i|\pi_i)$. At first, like Section 4.1, we assume $\pi_i = (\pi_{i,1}, \ldots, \pi_{i,M})$ and define $\psi(\pi_i) = (\frac{e^{\pi_{i,1}}}{\sum_{j=1}^{M} e^{\pi_{i,j}}}, \ldots, \frac{e^{\pi_{i,M}}}{\sum_{j=1}^{M} e^{\pi_{i,j}}})$ as the softmax function. Then, for a specific assignment $\sigma \in B_N(\mathbf{m}_i)$, we have that

$$\Pr\left(\{\mathbf{a}_{i,j} = \mathbf{e}_{\sigma(j)}\}_{j=1}^{N}|\pi_i\right) = \prod_{j=1}^{N} \frac{[\psi(\pi_i)]_{\sigma(j)}}{1 - \sum_{a=1}^{j-1} [\psi(\pi_i)]_{\sigma(a)}}, \tag{15}$$

where the symbol 'Pr' denotes the probability and $[\psi(\pi_i)]_j$ denotes its $j$-th component. Moreover, in Eq.15, when $j = 1$, we define the summation $\sum_{a=1}^{0} [\psi(\pi_i)]_{\sigma(a)} \equiv 0$ and simultaneously specify $\frac{0}{0} := 1$.

It is important to note that in Eq.15, the value $\frac{[\psi(\pi_i)]_{\sigma(j)}}{1 - \sum_{a=1}^{j-1} [\psi(\pi_i)]_{\sigma(a)}}$ stands for the $j$-step sampling-without-replacement probability. Finally, from the result of Eq.15, we have that

$$p(\mathbf{m}_i|\pi_i) = \sum_{\sigma \in B_N(\mathbf{m}_i)} \Pr\left(\{\mathbf{a}_{i,j} = \mathbf{e}_{\sigma(j)}\}_{j=1}^{N}|\pi_i\right)$$

$$= \sum_{\sigma \in B_N(\mathbf{m}_i)} \left(\prod_{j=1}^{N} \frac{[\psi(\pi_i)]_{\sigma(j)}}{1 - \sum_{a=1}^{j-1} [\psi(\pi_i)]_{\sigma(a)}}\right). \tag{16}$$

## B.2 Detailed Description of $p(\mathbf{m}|\pi)$

Due to that $p(\mathbf{m}|\pi) := \prod_{i=1}^{\frac{d}{M}} p(\mathbf{m}_i|\pi_i)$ and Eq.16, we then can show that

$$p(\mathbf{m}|\pi) := \prod_{i=1}^{\frac{d}{M}} p(\mathbf{m}_i|\pi_i) = \prod_{i=1}^{\frac{d}{M}} \left(\sum_{\sigma \in B_N(\mathbf{m}_i)} \left(\prod_{j=1}^{N} \frac{[\psi(\pi_i)]_{\sigma(j)}}{1 - \sum_{a=1}^{j-1} [\psi(\pi_i)]_{\sigma(a)}}\right)\right). \tag{17}$$

## B.3 Compute the Gradient $\nabla_\pi \log\left(p(\mathbf{m}|\pi)\right)$

Note that in Eq.10, in order to update the logits $\pi$ via mini-batch stochastic policy gradient descent, we need to frequently compute the gradient $\nabla_\pi \log\left(p(\mathbf{m}|\pi)\right)$. Thus, in this subsection, we give the detailed form of this $\nabla_\pi \log\left(p(\mathbf{m}|\pi)\right)$.

At first, due to that $p(\mathbf{m}|\pi) := \prod_{i=1}^{\frac{d}{M}} p(\mathbf{m}_i|\pi_i)$, we can know $\log\left(p(\mathbf{m}|\pi)\right) = \sum_{i=1}^{\frac{d}{M}} \log\left(p(\mathbf{m}_i|\pi_i)\right)$ such that

$$\nabla_\pi \log\left(p(\mathbf{m}|\pi)\right) = \left(\nabla_{\pi_1} \log\left(p(\mathbf{m}_1|\pi_1)\right), \nabla_{\pi_2} \log\left(p(\mathbf{m}_2|\pi_2)\right), \ldots, \nabla_{\pi_{\frac{d}{M}}} \log\left(p(\mathbf{m}_{\frac{d}{M}}|\pi_{\frac{d}{M}})\right)\right).$$

Therefore, in the subsequent part of this subsection, we show the specific form of $\nabla_{\pi_i} \log\left(p(\mathbf{m}_i|\pi_i)\right)$ for any $i \in [\frac{d}{M}]$. Like Section B.1, we assume that $\mathbf{m}_i = \sum_{i \in [N]} \mathbf{e}_{k_i}$ where $k_i \in [M], \forall i \in [N]$ and $k_1 \neq k_2 \neq \ldots \neq k_N$.

Then, when $\pi_i = (\pi_{i,1}, \ldots, \pi_{i,M})$, for any $k \in [M] = \{1, \ldots, M\}$, we have that

$$
\begin{aligned}
\frac{\partial \Big( \log \big( p(\mathbf{m}_i | \pi_i) \big) \Big)}{\partial \pi_{i,k}} &= \frac{1}{p(\mathbf{m}_i | \pi_i)} \frac{\partial \Big( p\left(\mathbf{m}_i | \pi_i\right) \Big)}{\partial \pi_{i,k}} \\
&= \frac{1}{p(\mathbf{m}_i | \pi_i)} \frac{\partial \left( \sum_{\sigma \in B_N(\mathbf{m}_i)} \left( \prod_{j=1}^N \frac{[\psi(\pi_i)]_{\sigma(j)}}{1 - \sum_{a=1}^{j-1} [\psi(\pi_i)]_{\sigma(a)}} \right) \right)}{\partial \pi_{i,k}} \\
&= \frac{1}{p(\mathbf{m}_i | \pi_i)} \sum_{\sigma \in B_N(\mathbf{m}_i)} \frac{\partial \left( \prod_{j=1}^N \frac{[\psi(\pi_i)]_{\sigma(j)}}{1 - \sum_{a=1}^{j-1} [\psi(\pi_i)]_{\sigma(a)}} \right)}{\partial \pi_{i,k}} \\
&= \frac{1}{p(\mathbf{m}_i | \pi_i)} \sum_{\sigma \in B_N(\mathbf{m}_i)} \left( \prod_{j=1}^N \frac{1}{1 - \sum_{a=1}^{j-1} [\psi(\pi_i)]_{\sigma(a)}} \right) \frac{\partial \left( \prod_{j=1}^N [\psi(\pi_i)]_{\sigma(j)} \right)}{\partial \pi_{i,k}} \\
&+ \frac{1}{p(\mathbf{m}_i | \pi_i)} \sum_{\sigma \in B_N(\mathbf{m}_i)} \left( \prod_{j=1}^N [\psi(\pi_i)]_{\sigma(j)} \right) \frac{\partial \left( \prod_{j=1}^N \frac{1}{1 - \sum_{a=1}^{j-1} [\psi(\pi_i)]_{\sigma(a)}} \right)}{\partial \pi_{i,k}}.
\end{aligned}
\tag{18}
$$

Next, we compute the $\frac{\partial \left( \prod_{j=1}^N [\psi(\pi_i)]_{\sigma(j)} \right)}{\partial \pi_{i,k}}$ and $\frac{\partial \left( \prod_{j=1}^N \frac{1}{1 - \sum_{a=1}^{j-1} [\psi(\pi_i)]_{\sigma(a)}} \right)}{\partial \pi_{i,l}}$ in Eq.18. At first, from the definition of $\psi(\pi_i)$, we can show

$$
\begin{aligned}
\frac{\partial [\psi(\pi_i)]_j}{\partial \pi_{i,k}} &= - [\psi(\pi_i)]_k * [\psi(\pi_i)]_j \, , \forall k \neq j \in [M]; \\
\frac{\partial [\psi(\pi_i)]_k}{\partial \pi_{i,k}} &= [\psi(\pi_i)]_k \left( 1 - [\psi(\pi_i)]_k \right).
\end{aligned}
\tag{19}
$$

As a result, we have that

$$
\frac{\partial \left( \prod_{j=1}^N [\psi(\pi_i)]_{\sigma(j)} \right)}{\partial \pi_{i,k}} = \left( \prod_{j=1}^N [\psi(\pi_i)]_{\sigma(j)} \right) \left( \mathbb{I}\left[ k \in \{k_i\}_{i=1}^N \right] - N * [\psi(\pi_i)]_k \right),
\tag{20}
$$

where $\mathbb{I}$ is the indicator function.

As for $\frac{\partial \left( \prod_{j=1}^N \frac{1}{1 - \sum_{a=1}^{j-1} [\psi(\pi_i)]_{\sigma(a)}} \right)}{\partial \pi_{i,l}}$, we have that

$$
\begin{aligned}
&\frac{\partial \left( \prod_{j=1}^N \frac{1}{1 - \sum_{a=1}^{j-1} [\psi(\pi_i)]_{\sigma(a)}} \right)}{\partial \pi_{i,k}} \\
&= \left( \prod_{j=1}^N \frac{1}{1 - \sum_{a=1}^{j-1} [\psi(\pi_i)]_{\sigma(a)}} \right) \left( \sum_{j=1}^N \frac{\frac{\partial \left( \sum_{a=1}^{j-1} [\psi(\pi_i)]_{\sigma(a)} \right)}{\partial \pi_{i,k}}}{1 - \sum_{a=1}^{j-1} [\psi(\pi_i)]_{\sigma(a)}} \right) \\
&= \left( \prod_{j=1}^N \frac{1}{1 - \sum_{a=1}^{j-1} [\psi(\pi_i)]_{\sigma(a)}} \right) \left( \sum_{j=1}^N \frac{[\psi(\pi_i)]_k \left( \mathbb{I}[j > \sigma^{-1}(k)] - \sum_{a=1}^{j-1} [\psi(\pi_i)]_{\sigma(a)} \right)}{1 - \sum_{a=1}^{j-1} [\psi(\pi_i)]_{\sigma(a)}} \right),
\end{aligned}
\tag{21}
$$

where $\mathbb{I}$ is the indicator function and $\sigma^{-1}$ denotes the inverse mapping of $\sigma$. Especially when $k \in \{k_1, \ldots, k_N\}$, e.g., $k = k_c$ where $c \in [N]$, we set $\sigma^{-1}(k) = c$. As for $k \neq \{k_1, \ldots, k_N\}$, we set $\sigma^{-1}(k) = \infty$ such that $\mathbb{I}[j > \sigma^{-1}(k)] = 0$ for any $j \in [N]$.

Merging Eq.21 and Eq.20 into Eq.18, we can finally have that

$$\frac{\partial\Big(\log\left(p(\mathbf{m}_i|\pi_i)\right)\Big)}{\partial\pi_{i,k}} = \sum_{\sigma\in B_N(\mathbf{m}_i)}\left(\prod_{j=1}^{N}\frac{[\psi(\pi_i)]_{\sigma(j)}}{1-\sum_{a=1}^{j-1}[\psi(\pi_i)]_{\sigma(a)}}\right)\frac{\Big(\mathbb{I}\left[k\in\{k_i\}_{i=1}^{N}\right]-N\,[\psi(\pi_i)]_k\Big)}{p(\mathbf{m}_i|\pi_i)}$$

$$+\sum_{\sigma\in B_N}\left(\prod_{j=1}^{N}\frac{[\psi(\pi_i)]_{\sigma(j)}}{1-\sum_{a=1}^{j-1}[\psi(\pi_i)]_{\sigma(a)}}\right)\left(\sum_{j=1}^{N}\frac{[\psi(\pi_i)]_k\Big(\mathbb{I}[j>\sigma^{-1}(k)]-\sum_{a=1}^{j-1}[\psi(\pi_i)]_{\sigma(a)}\Big)}{p(\mathbf{m}_i|\pi_i)\Big(1-\sum_{a=1}^{j-1}[\psi(\pi_i)]_{\sigma(a)}\Big)}\right),$$

where $p(\mathbf{m}_i|\pi_i)=\sum_{\sigma\in B_N(\mathbf{m}_i)}\left(\prod_{j=1}^{N}\frac{[\psi(\pi_i)]_{\sigma(j)}}{1-\sum_{i=1}^{j-1}[\psi(\pi_i)]_{\sigma(i)}}\right)$ and $\mathbf{m}_i=\sum_{i\in[N]}\mathbf{e}_{k_i}\in\mathcal{S}^{N:M}$.

## C  PROOFS

In this Section, we provide the detailed proofs of the main theorems.

### C.1  PROOF OF THEOREM 1

This subsection aims to present a rigorous proof for the representation Theorem 1. Before going in the details, we first assume that, in Eq.4, $\mathbf{a}_i = \mathbf{e}_{k_i}, \forall i \in [N]$ where $k_i \in [M], \forall i \in [N]$ and $k_1 \neq k_2 \neq \ldots \neq k_N$. With this assumption, then we can show that,

$$\bigodot_{i=1}^{N}(\mathbf{1}_M - \mathbf{a}_i) = \mathbf{1}_M - \left(\sum_{j\in\{k_1,\ldots,k_N\}}\mathbf{e}_j\right). \tag{22}$$

We verify this Eq.22 by induction. Firstly, when $N = 1$, Eq. 22 naturally holds. Subsequently, we assume that when $N = m < M$, Eq. 22 is right. As a result, we can show that, when $N = m+1 \leq M$

$$\bigodot_{i=1}^{N}(\mathbf{1}_M - \mathbf{a}_i) = \bigodot_{i=1}^{m+1}(\mathbf{1}_M - \mathbf{a}_i) = \left(\bigodot_{i=1}^{m}(\mathbf{1}_M - \mathbf{a}_i)\right)\odot(\mathbf{1}_M - \mathbf{a}_{m+1})$$

$$= \left(\mathbf{1}_M - \sum_{j\in\{k_1,\ldots,k_m\}}\mathbf{e}_j\right)\odot\left(\mathbf{1}_M - \mathbf{e}_{k_{m+1}}\right)$$

$$= \mathbf{1}_M - \left(\sum_{j\in\{k_1,\ldots,k_m\}}\mathbf{e}_j\right) - \mathbf{e}_{k_{m+1}} - \sum_{j\in\{k_1,\ldots,k_m\}}\left(\mathbf{e}_{k_{m+1}}\odot\mathbf{e}_j\right) = \mathbf{1}_M - \left(\sum_{j\in\{k_1,\ldots,k_{m+1}\}}\mathbf{e}_j\right),$$

where the final equality follows from that $\mathbf{e}_{k_{m+1}}\odot\mathbf{e}_j = 0$, when $j \neq k_{m+1}$. As a result, the Eq. 22 holds for any $N \leq M$.

According to the result of Eq. 22, we can easily have that

$$\bigoplus_{i=1}^{N}\mathbf{a}_i = \mathbf{1} - \bigodot_{i=1}^{N}\left(\mathbf{1} - \mathbf{a}_i\right) = \sum_{j\in\{k_1,\ldots,k_N\}}\mathbf{e}_j. \tag{23}$$

Therefore, from Eq.23, we can infer that, when $\mathbf{a}_i \in \{\mathbf{e}_1,\ldots,\mathbf{e}_M\}, \forall i \in [N]$ and $\mathbf{a}_1 \neq \mathbf{a}_2 \neq \ldots \neq \mathbf{a}_N$, $\bigoplus_{i=1}^{N}\mathbf{a}_i \in \mathbb{B}^{1\times M}$ and $\|\bigoplus_{i=1}^{N}\mathbf{a}_i\|_1 = N$ such that $\bigoplus_{i=1}^{N}\mathbf{a}_i \in \mathcal{S}^{N:M}$ where $\mathbb{B}$ denotes the Boolean set. Furthermore, for any binary vector $\mathbf{b} \in \mathbb{M}^{N:M}$, we can redefine $\mathbf{b} = \sum_{i\in[N]}\mathbf{e}_{s_i}$ where $s_i \in [M], \forall i \in [N]$ and $s_1 \neq s_2 \neq \ldots \neq s_N$. Then, if we set $\mathbf{a}_i = \mathbf{e}_{s_i}$ for $i \in \{1,\ldots,n\}$, acoording to the result of Eq.23, we can have

$$\bigoplus_{i=1}^{N}\mathbf{a}_i = \sum_{i\in[N]}\mathbf{e}_{s_i} = \mathbf{b}.$$

As a result, we can establish that

$$\mathcal{S}^{N:M} = \left\{\bigoplus_{i=1}^{N}\mathbf{a}_i : \mathbf{a}_i \in \{\mathbf{e}_1,\ldots,\mathbf{e}_M\}, \forall i \in [N] \text{ and } \mathbf{a}_1 \neq \mathbf{a}_2 \neq \ldots \neq \mathbf{a}_N\right\}.$$

## C.2 PROOF OF POLICY GRADIENT ESTIMATOR EQUATION 9

From the notations in Eq.8, we have that

$$\Phi(\pi) := \mathbb{E}_{\xi \sim \mathcal{D}, \mathbf{m}=\{\mathbf{m}_i | \mathbf{m}_i \sim p(\mathbf{m}_i | \pi_i)\}} \left[ f\left(\mathbf{m} \odot \mathbf{w}, \xi\right) \right] := \int \mathbb{E}_{\xi} \left[ f\left(\mathbf{m} \odot \mathbf{w}, \xi\right) \right] p(\mathbf{m}|\pi) \mathrm{d}\mathbf{m}. \quad (24)$$

It is worth noting that in right-hand side(RHS) of Eq.24, only the component "$p(\mathbf{m}|\pi)$" contains the unknown logits variable $\pi$. As a result, we have that

$$\nabla_\pi \Phi(\pi) = \nabla_\pi \int \mathbb{E}_\xi \left[ f\left(\mathbf{m} \odot \mathbf{w}, \xi\right) \right] p(\mathbf{m}|\pi) \mathrm{d}\mathbf{m}$$

$$= \int \mathbb{E}_\xi \left[ f\left(\mathbf{m} \odot \mathbf{w}, \xi\right) \right] \nabla_\pi p(\mathbf{m}|\pi) \mathrm{d}\mathbf{m}$$

$$= \int \mathbb{E}_\xi \left[ f\left(\mathbf{m} \odot \mathbf{w}, \xi\right) \right] \frac{\nabla_\pi p(\mathbf{m}|\pi)}{p(\mathbf{m}|\pi)} p(\mathbf{m}|\pi) \mathrm{d}\mathbf{m}$$

$$= \int \mathbb{E}_\xi \left[ f\left(\mathbf{m} \odot \mathbf{w}, \xi\right) \right] \left( \nabla_\pi \log\left(p(\mathbf{m}|\pi)\right) \right) p(\mathbf{m}|\pi) \mathrm{d}\mathbf{m}$$

$$= \mathbb{E}_{\xi \sim \mathcal{D}, \mathbf{m}=\{\mathbf{m}_i | \mathbf{m}_i \sim p(\mathbf{m}_i | \pi_i)\}} \left[ f(\mathbf{m} \odot \mathbf{w}, \xi) \nabla \log\left(p(\mathbf{m}|\pi)\right) \right],$$

where the forth equality comes from the relationship that $\left( \log(f(x)) \right)' = \frac{\mathrm{d}\left( \log(f(x)) \right)}{\mathrm{d}x} = \frac{f'(x)}{f(x)}$.

## C.3 PROOF OF THEOREM 2

We first investigate the properties of the policy gradient update method applied in this paper. As shown in Eq.(9), the general policy gradient satisfies the following equation:

$$\nabla \Phi(\pi) = \mathbb{E}_{p(\mathbf{m}|\pi)} \left[ f(\mathbf{m} \odot \mathbf{w}) \nabla \log\left(p(\mathbf{m}|\pi)\right) \right],$$

where $\mathbb{E}_\xi \left[ f(\mathbf{m} \odot \mathbf{w}, \xi) \right] = f(\mathbf{m} \odot \mathbf{w})$.

In the training, due to the limitations of data samples, instead of computing the full loss $f(\mathbf{m} \odot \mathbf{w})$, we typically use a small mini-batch stochastic gradient, that is,

$$g_\mathrm{p} = f(\mathbf{m} \odot \mathbf{w}, \xi) \nabla \log\left(p(\mathbf{m}|\pi)\right),$$

### C.3.1 LOSS RESIDUAL AND SMOOTHING TRACKER ARE UNBIASED ESTIMATORS OF $\nabla\Phi(\pi)$

We denote $g_\mathrm{r}$ as update via loss residual:

$$g_\mathrm{r} = \left( f(\mathbf{m} \odot \mathbf{w}, \xi) - f(\mathbf{m}_0 \odot \mathbf{w}, \xi) \right) \nabla \log\left(p(\mathbf{m}|\pi)\right),$$

and $g_\mathrm{sr}$ as update via loss residual with smoothing tracker:

$$g_\mathrm{sr} = \left( f(\mathbf{m} \odot \mathbf{w}, \xi) - f(\mathbf{m}_0 \odot \mathbf{w}, \xi) - \delta \right) \nabla \log\left(p(\mathbf{m}|\pi)\right),$$
$$\delta = \alpha \delta + (1 - \alpha) \left( f(\mathbf{m} \odot \mathbf{w}, \xi) - f(\mathbf{m}_0 \odot \mathbf{w}, \xi) \right).$$

It is worth noting that these two introduced additional terms $f(\mathbf{m}_0 \odot \mathbf{w}, \xi)$ and $\delta$ are independent of the logits variable $\pi$ such that we can know that

$$\mathbb{E}_{p(\mathbf{m}|\pi)} \left[ \left( f(\mathbf{m}_0 \odot \mathbf{w}, \xi) + \delta \right) \nabla \log\left(p(\mathbf{m}|\pi)\right) \right] = \left( f(\mathbf{m}_0 \odot \mathbf{w}, \xi) + \delta \right) \int p(\mathbf{m}|\pi) \frac{\nabla p(\mathbf{m}|\pi)}{p(\mathbf{m}|\pi)} d\mathbf{m}$$

$$= \left( f(\mathbf{m}_0 \odot \mathbf{w}, \xi) + \delta \right) \nabla \int p(\mathbf{m}|\pi) d\mathbf{m} = \left( f(\mathbf{m}_0 \odot \mathbf{w}, \xi) + \delta \right) \nabla 1 = 0.$$

Therefore, our proposed update using the loss residual with smoothing tracker remains an unbiased estimator of the standard policy gradient. Similarly, letting $\delta = 0$, it degrades to the update with only loss residual, which is also a unbiased estimator of the standard policy gradient. In fact, our proposed enhanced version of the policy gradient update can be viewed as an auxiliary training method that introduces a baseline term, similar to the approach in reinforcement learning (Williams, 1992). Results are also consistent with existing similar analysis (Fan et al., 2025c).

### C.3.2 Efficiency of $g_{\mathrm{R}}$

We first investigate the properties of updating via loss residual $f(\mathbf{m} \odot \mathbf{w}, \xi) - f(\mathbf{m}_0 \odot \mathbf{w}, \xi)$. We have the variance of the standard policy gradient $g_{\mathrm{p}}$ as:

$$\mathrm{Var}\left[g_{\mathrm{p}}\right] = \mathbb{E}\left[f(\mathbf{m} \odot \mathbf{w}, \xi)^2 \left(\nabla \log\left(p(\mathbf{m}|\pi)\right)\right)^2\right] - \mathbb{E}\left[f(\mathbf{m} \odot \mathbf{w}, \xi) \nabla \log\left(p(\mathbf{m}|\pi)\right)\right]^2$$

$$= \mathbb{E}\left[\left(f(\mathbf{m} \odot \mathbf{w}, \xi)\right)^2 \left(\nabla \log\left(p(\mathbf{m}|\pi)\right)\right)^2\right] - \nabla\Phi(\pi)^2.$$

Similarly, since $\mathbb{E}\left[g_{\mathrm{r}}\right] = \nabla\Phi(\pi)$, the variance of $g_{\mathrm{r}}$ achieves:

$$\mathrm{Var}\left[g_{\mathrm{r}}\right] = \mathbb{E}\left[\left(f(\mathbf{m} \odot \mathbf{w}, \xi) - f(\mathbf{m}_0 \odot \mathbf{w}, \xi)\right)^2 \left(\nabla \log\left(p(\mathbf{m}|\pi)\right)\right)^2\right] - \nabla\Phi(\pi)^2.$$

Thus we have:

$$\mathrm{Var}\left[g_{\mathrm{r}}\right] - \mathrm{Var}\left[g_{\mathrm{p}}\right]$$

$$= \mathbb{E}_{p(\mathbf{m}|\pi)}\left[\left(\nabla \log\left(p(\mathbf{m}|\pi)\right)\right)^2 \mathbb{E}_{\xi}\left[\left(f(\mathbf{m} \odot \mathbf{w}, \xi) - f(\mathbf{m}_0 \odot \mathbf{w}, \xi)\right)^2 - f(\mathbf{m} \odot \mathbf{w}, \xi)^2\right]\right]$$

$$= \mathbb{E}_{p(\mathbf{m}|\pi)}\left[\underbrace{\left(\nabla \log\left(p(\mathbf{m}|\pi)\right)\right)^2}_{\geq 0} \mathbb{E}_{\xi}\left[\underbrace{f(\mathbf{m}_0 \odot \mathbf{w}, \xi)}_{\geq 0}\left(f(\mathbf{m}_0 \odot \mathbf{w}, \xi) - 2f(\mathbf{m} \odot \mathbf{w}, \xi)\right)\right]\right].$$

Their relative magnitudes are determined by $f(\mathbf{m}_0 \odot \mathbf{w}, \xi) - 2f(\mathbf{m} \odot \mathbf{w}, \xi)$ term and we have:

$$\begin{cases} \mathrm{Var}\left[g_{\mathrm{r}}\right] \geq \mathrm{Var}\left[g_{\mathrm{p}}\right], & \text{when} \quad f(\mathbf{m}_0 \odot \mathbf{w}, \xi) \geq 2f(\mathbf{m} \odot \mathbf{w}, \xi), \\ \mathrm{Var}\left[g_{\mathrm{r}}\right] < \mathrm{Var}\left[g_{\mathrm{p}}\right], & \text{when} \quad f(\mathbf{m}_0 \odot \mathbf{w}, \xi) < 2f(\mathbf{m} \odot \mathbf{w}, \xi). \end{cases}$$

Therefore, updating via loss residual can always achieve a lower variance when $f(\mathbf{m} \odot \mathbf{w}, \xi) > \frac{1}{2}f(\mathbf{m}_0 \odot \mathbf{w}, \xi)$. This implies that the variance in the initial training stage is significantly lower than that of the vanilla PGE $g_{\mathrm{p}}$, enabling substantial acceleration. We also validate this in our experiments, where the vanilla policy gradient converges extremely slowly and barely learns effective information, while $g_{\mathrm{r}}$ can achieve a rapid reduction in loss within only hundreds of iterations.

### C.3.3 Efficiency of $g_{\mathrm{SR}}$

Similarly, since $\mathbb{E}\left[g_{\mathrm{sr}}\right] = \nabla\Phi(\pi)$, the variance of $g_{\mathrm{sr}}$ achieves:

$$\mathrm{Var}\left[g_{\mathrm{sr}}\right] = \mathbb{E}\left[\left(f(\mathbf{m} \odot \mathbf{w}, \xi) - f(\mathbf{m}_0 \odot \mathbf{w}, \xi) - \delta\right)^2 \left(\nabla \log\left(p(\mathbf{m}|\pi)\right)\right)^2\right] - \nabla\Phi(\pi)^2.$$

And we have:

$$\mathrm{Var}\left[g_{\mathrm{sr}}\right] - \mathrm{Var}\left[g_{\mathrm{p}}\right]$$

$$= \mathbb{E}_{p(\mathbf{m}|\pi)}\left[\left(\nabla \log\left(p(\mathbf{m}|\pi)\right)\right)^2 \mathbb{E}_{\xi}\left[\left(f(\mathbf{m} \odot \mathbf{w}, \xi) - f(\mathbf{m}_0 \odot \mathbf{w}, \xi) - \delta\right)^2 - f(\mathbf{m} \odot \mathbf{w}, \xi)^2\right]\right]$$

$$= \mathbb{E}_{p(\mathbf{m}|\pi)}\left[\left(\nabla \log\left(p(\mathbf{m}|\pi)\right)\right)^2 \mathbb{E}_{\xi}\left[f(\mathbf{m}_0 \odot \mathbf{w}, \xi)^2\right]\right] + \mathbb{E}_{p(\mathbf{m}|\pi)}\left[\left(\nabla \log\left(p(\mathbf{m}|\pi)\right)\right)^2 \mathbb{E}_{\xi}\left[\delta^2\right]\right]$$

$$\quad - 2\mathbb{E}_{p(\mathbf{m}|\pi)}\left[\left(\nabla \log\left(p(\mathbf{m}|\pi)\right)\right)^2 \mathbb{E}_{\xi}\left[f(\mathbf{m} \odot \mathbf{w}, \xi)\delta\right]\right]$$

$$\quad + 2\mathbb{E}_{p(\mathbf{m}|\pi)}\left[\left(\nabla \log\left(p(\mathbf{m}|\pi)\right)\right)^2 \mathbb{E}_{\xi}\left[f(\mathbf{m}_0 \odot \mathbf{w}, \xi)\delta\right]\right]$$

$$\quad - 2\mathbb{E}_{p(\mathbf{m}|\pi)}\left[\left(\nabla \log\left(p(\mathbf{m}|\pi)\right)\right)^2 \mathbb{E}_{\xi}\left[f(\mathbf{m} \odot \mathbf{w}, \xi)f(\mathbf{m}_0 \odot \mathbf{w}, \xi)\right]\right]$$

$$= \underbrace{\mathbb{E}_{p(\mathbf{m}|\pi)}\left[\left(\nabla \log\left(p(\mathbf{m}|\pi)\right)\right)^2\right]\delta^2 + 2\mathbb{E}_{p(\mathbf{m}|\pi)}\left[\left(\nabla \log\left(p(\mathbf{m}|\pi)\right)\right)^2\left(f(\mathbf{m}_0 \odot \mathbf{w}) - f(\mathbf{m} \odot \mathbf{w})\right)\right]\delta}_{\text{denoted by } A \geq 0 \qquad\qquad\qquad \text{denoted by } B}$$

$$\underbrace{+ \mathbb{E}_{p(\mathbf{m}|\pi)}\left[\left(\nabla \log\left(p(\mathbf{m}|\pi)\right)\right)^2 \mathbb{E}_{\xi}\left[f(\mathbf{m}_0 \odot \mathbf{w}, \xi)\left(f(\mathbf{m}_0 \odot \mathbf{w}, \xi) - 2f(\mathbf{m} \odot \mathbf{w}, \xi)\right)\right]\right]}_{\mathrm{Var}[g_r] - \mathrm{Var}[g_p]}.$$

Clearly, when $\delta = 0$, $\text{Var}\,[g_{\text{sr}}] = \text{Var}\,[g_{\text{r}}]$. Next, we discuss the case where $\delta \neq 0$. The above expression can be viewed as a quadratic function w.r.t. $\delta$, i.e.,

$$\text{Var}\,[g_{\text{sr}}] - \text{Var}\,[g_{\text{p}}] = V(\delta) = A\delta^2 + B\delta + (\text{Var}\,[g_{\text{r}}] - \text{Var}\,[g_{\text{p}}]),$$

According to the definition of $\delta$, it is the moving average of the $f(\mathbf{m} \odot \mathbf{w}, \xi) - f(\mathbf{m}_0 \odot \mathbf{w}, \xi)$ term. By considering $f(\mathbf{m} \odot \mathbf{w}, \xi) \geq \frac{1}{2}f(\mathbf{m}_0 \odot \mathbf{w}, \xi)$, we can intuitively examine the corresponding magnitude relationships through the function plots. As shown in Figure 7, when $|\delta| < |\frac{B}{A}|$, we always have $\text{Var}\,[g_{\text{sr}}] < \text{Var}\,[g_{\text{r}}]$. Furthermore, if $\delta = -\frac{B}{2A}$, the extent of variance reduction will reach its maximum. Therefore we have:

$$\delta^\star = -\frac{B}{2A} = \frac{\mathbb{E}_{p(\mathbf{m}|\pi)}\left[(\nabla \log (p(\mathbf{m}|\pi)))^2 (f(\mathbf{m} \odot \mathbf{w}) - f(\mathbf{m}_0 \odot \mathbf{w}))\right]}{\mathbb{E}_{p(\mathbf{m}|\pi)}\left[(\nabla \log (p(\mathbf{m}|\pi)))^2\right]}$$

$$= \mathbb{E}_{p(\mathbf{m}|\pi)}\left[\frac{(\nabla \log (p(\mathbf{m}|\pi)))^2}{\mathbb{E}_{p(\mathbf{m}|\pi)}\left[(\nabla \log (p(\mathbf{m}|\pi)))^2\right]} (f(\mathbf{m} \odot \mathbf{w}) - f(\mathbf{m}_0 \odot \mathbf{w}))\right]$$

$$= \mathbb{E}_{\hat{p}(\mathbf{m}|\pi)}\left[f(\mathbf{m} \odot \mathbf{w}) - f(\mathbf{m}_0 \odot \mathbf{w})\right],$$

where $\hat{p}(\mathbf{m}|\pi) = \frac{p(\mathbf{m}|\pi)(\nabla \log(p(\mathbf{m}|\pi)))^2}{\mathbb{E}_{p(\mathbf{m}|\pi)}\left[(\nabla \log(p(\mathbf{m}|\pi)))^2\right]}$.

Clearly, $\delta^\star$ can be interpreted as the expectation of $f(\mathbf{m} \odot \mathbf{w}) - f(\mathbf{m}_0 \odot \mathbf{w})$ under the optimal distribution $\hat{p}(\mathbf{m}|\pi)$, or equivalently, as the weighted average over all possible cases. It is feasible to accurately measure this distribution. When the original distribution $p(\mathbf{m}|\pi)$ is known, the optimal distribution can be derived; however, the corresponding computational overhead to calculate it is prohibitively high. Therefore, we track all stochastic sampling in the training process and calculate the moving average of each $f(\mathbf{m}_t \odot \mathbf{w}, \xi) - f(\mathbf{m}_0 \odot \mathbf{w}, \xi)$ as a compromise. After a long iteration $t$ and enough samplings, $\delta$ can achieve significant and stable performance.

Therefore, we have $\text{Var}\,[g_{\text{sr}}] \lesssim \text{Var}\,[g_{\text{r}}] < \text{Var}\,[g_{\text{p}}]$.

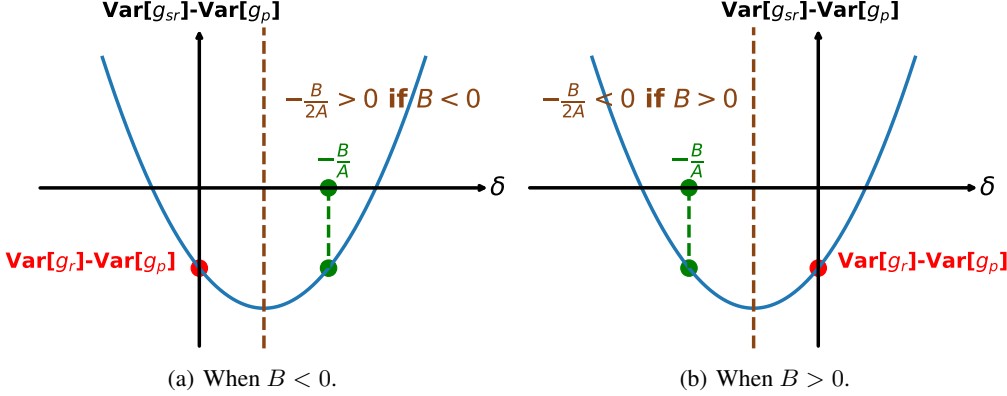

(a) When $B < 0$.          (b) When $B > 0$.

Figure 7: Schematic illustration of the optimal solution.

And the theoretically maximal variance reduction can be expressed as:

$$
\max \left\{ \text{Var}\left[g_{\text{p}}\right] - \text{Var}\left[g_{\text{sr}}\right] \right\}
$$

$$
= -\mathbb{E}_{p(\mathbf{m}|\pi)} \left[ \left(\nabla \log\left(p(\mathbf{m}|\pi)\right)\right)^2 \mathbb{E}_{\xi}\left[ f(\mathbf{m}_0 \odot \mathbf{w}, \xi)\left( f(\mathbf{m}_0 \odot \mathbf{w}, \xi) - 2f(\mathbf{m} \odot \mathbf{w}, \xi)\right)\right] \right]
$$

$$
+ \frac{\left( \mathbb{E}_{p(\mathbf{m}|\pi)} \left[ \left(\nabla \log\left(p(\mathbf{m}|\pi)\right)\right)^2 \left(f(\mathbf{m} \odot \mathbf{w}) - f(\mathbf{m}_0 \odot \mathbf{w})\right)\right]\right)^2}{\mathbb{E}_{p(\mathbf{m}|\pi)}\left[ \left(\nabla \log\left(p(\mathbf{m}|\pi)\right)\right)^2\right]}
$$

$$
= \text{Var}\left[g_{\text{p}}\right] - \text{Var}\left[g_{\text{r}}\right] + \frac{\left( \mathbb{E}_{p(\mathbf{m}|\pi)} \left[ \left(\nabla \log\left(p(\mathbf{m}|\pi)\right)\right)^2 \left(f(\mathbf{m} \odot \mathbf{w}) - f(\mathbf{m}_0 \odot \mathbf{w})\right)\right]\right)^2}{\mathbb{E}_{p(\mathbf{m}|\pi)}\left[ \left(\nabla \log\left(p(\mathbf{m}|\pi)\right)\right)^2\right]}.
$$

