# OpenReview forum: "MaskPro: Linear-Space Probabilistic Learning for Strict (N:M)-Sparsity on LLMs"
_ICLR.cc/2026/Conference — ICLR 2026 Poster_

### Official Review · Reviewer_HgWG · 2025-10-28

**Soundness:** 3
**Presentation:** 3
**Contribution:** 2
**Rating:** 6
**Confidence:** 3

**Summary:**

The paper proposes **MaskPro**, a probabilistic framework to learn strict (N:M)-sparsity masks for LLM weights with **linear memory** in the number of parameters. It models each group of M weights with a categorical distribution and generates N active positions via sampling without replacement. It trains logits using a policy-gradient estimator on loss residuals with a moving average baseline to cut variance. A representation theorem rewrites N:M masks as a probabilistic sum of basis vectors; proofs claim unbiased gradients and reduced variance. Experiments on several 7B models show competitive zero-shot accuracy vs rule-based pruning and proximity to MaskLLM at much lower memory and data cost.

**Strengths:**

- **Memory scalability.** Reduces logits storage from combinatorial $O \left(\binom{M}{N}^{d/M}\right)$ to linear $O(d)$. This is the key lever for practicality.
- **Clean probabilistic formulation.** N-way sampling without replacement per group with explicit $p(m\mid \pi)$ and REINFORCE-style training.
- **Variance control.** Loss-residual update with a moving average tracker yields an unbiased estimator with lower variance than vanilla PG under stated conditions. Empirical curves support stability gains.
- **Robustness to tiny data.** Reasonable performance even with very few calibration samples; plots include the 1-sample case.
- **Empirical breadth.** Evaluates on multiple 7B models and diverse zero-shot tasks; MaskPro beats rule-based baselines by ~2 points on average and narrows the gap to MaskLLM.
- **Training efficiency.** Orders-of-magnitude lower memory and dataset size than MaskLLM; single-GPU feasibility reported.

**Weaknesses:**

- **Novelty vs known tools.** The representation of N:M masks via basis-vector sums and a categorical policy with REINFORCE is conceptually straightforward; the main novelty is the linear-space parameterization plus the residual baseline. The paper should better position against prior PG-based pruning and Gumbel/relaxation methods.
- **Sampling cost.** The approach still simulates N-way sampling per group at train time. The paper notes this as a time bottleneck but gives limited profiling or asymptotic-to-wall-clock mapping.
- **Baseline fairness and scope.** Results focus on (2:4) with (4:8) pushed to appendix. No strong hardware-level throughput benchmarks for inference with real N:M kernels. Comparisons use C4 for all methods, but details like per-layer masks and calibration choices could bias outcomes; MaskLLM numbers are partly taken from prior work.
- **Initialization dependence.** Logit init relies on a good starting mask (Top-N or SparseGPT). The sensitivity to poorer inits and to the magnitude hyperparameter is only partly explored.
- **Theory conditions.** The variance-reduction claim depends on a condition $f(m_t \odot w,\xi) > \tfrac12 f(m_0\odot w,\xi)$. Practical validity across models and tasks is not deeply tested; switching $m_0$ mid-training is suggested but not ablated.
- **Model scale.** Only 7B models are reported. No 13B–70B results, where memory and sampling costs and accuracy regressions are more consequential.

**Questions:**

1. How is $p(m\mid \pi)$ computed efficiently for N-way sampling without replacement? Provide explicit formulas and complexity, not just appendix references. Can you replace sampling with a differentiable top-N estimator while keeping linear memory?
2. How sensitive is performance to the initial mask $m_0$ and the logit magnitude $C$? Report ablations that start from random $m_0$, weaker heuristics, and different $C$.
3. Give wall-clock profiles: percent time in sampling, forward passes, and PG updates. Can cached logits or Gumbel-Top-k variants reduce sampling cost further?
4. Show end-to-end throughput and latency on A100/H100 with vendor N:M kernels before and after MaskPro, at fixed accuracy. Accuracy-only tables are insufficient to claim deployment readiness.

---

> ### Author Response · Authors · 2025-11-22
> **Response to Reviewer HgWG (1/4)**
>
> ## W1: The representation of N:M masks via basis-vector sums and a categorical policy with REINFORCE is conceptually straightforward; the main novelty is the linear-space parameterization plus the residual baseline. The paper should better position against prior PG-based pruning and Gumbel/relaxation methods.
> We sincerely thank the reviewer for raising this question.
>
> Our work is inspired by MaskLLM and further generalizes its ideas. To the best of our knowledge, MaskLLM is the first pioneering work that trains a strict (N:M) sparse mask end-to-end within LLMs, and it has made significant contributions to this area. However, due to its full modeling of the combinatorial space and reliance on gradient-based training, its experimental overhead is extremely high. Therefore, we aim to develop a resource-efficient training framework that further advances the trainability of strict (N:M) sparsity. Our core contributions include introducing a new sampling procedure that trades distributional transformation for reduced memory consumption on logits, and replacing gradient-based updates with a refined policy-gradient estimator to enable effective end-to-end training. We again thank the reviewer for the helpful suggestions. We have revised our main contribution to clarify and highlight these points.
>
>
> ## W2: The approach still simulates N-way sampling per group at train time. The paper notes this as a time bottleneck but gives limited profiling or asymptotic-to-wall-clock mapping.
> We sincerely thank the reviewer for raising this issue.
>
> We use torch.multinomial function for (N:M)-sparsity sampling, which simulates the sampling process through a lookup-based mechanism and provides high accuracy. The forward pass is implemented through a standard wrapper function. Specifically, we wrap the linear layer with an additional mask parameter and integrate the mask computation directly inside the linear operation. This design avoids modifying PyTorch's computation graph and enables efficient inference. The logits updates are computed entirely through matrix calculation, and PyTorch's built-in libraries already provide the necessary parallelization. To further illustrate the implementation details, we report the per-step training time as follows:
>
> |          | Mask Sampling | Ratio | Forward | Ratio | PG Update | Ratio |
> | :-: | :-: | :-: | :-: | :-: | :-: | :-: |
> | 7B  | 2.328s | 85.94% | 0.062s | 2.29% | 0.319s | 11.77% |
> | 13B | 4.739s | 85.83% | 0.139s | 2.52% | 0.644s | 11.65% |
>
> The main source of time consumption comes from the sampling process. We also evaluated the sampling performance under different sparsity patterns, as shown in the table below.
>
> |          | (2:4) | (4:8) | (8:16) |
> | :-: | :-: | :-: | :-: |
> | LLaMA-7B  | 2.328s | 1.334s | 0.794s |
> | LLaMA-13B | 4.739s | 2.692s | 1.574s |
>
> We can observe that doubling the model size roughly doubles the sampling time. Another interesting observation is that the sampling time of (N:M)-Sparsity depends on $M$. With the same model size, a larger $M$ leads to shorter sampling time, due to parallel optimizations in the sampling process. For a $d$-dimensional model, there are a total of $\frac{d}{M}$ sampling groups. Although increasing $N$ and $M$ makes each group more expensive to sample, the total number of parallel groups decreases proportionally. This reduction in the number of groups results in a more favorable computation pattern for hardware. Consequently, for more complex (N:M)-Sparsity, the time required for a single sampling step can actually be lower. **Large M is a GPU-friendly selection.**
>
> ## W3: Results focus on (2:4) with (4:8) pushed to appendix. No strong hardware-level throughput benchmarks for inference with real N:M kernels.
>
> We thank the reviewer for raising this technical question. Since 2:4 sparsity has already been extensively studied in the context of hardware acceleration, our paper primarily focuses on proposing an algorithmic framework for discovering semi-structured sparsity. The inference-time acceleration follows standard practices, and thus carries no additional methodological differences beyond existing sparse kernel implementations.
>
> To address the reviewer's concerns, we follow the methodology described in
> https://pytorch.org/blog/accelerating-neural-network-training/ to replace the dense linear layers with their sparse counterparts and evaluate the performance using sparse_GEMM. We additionally test both the fused and unfused versions of the sparse operators. The results are summarized below:
>
> | Dense | SemiSparseLinear | Compile | Time |
> | :-: | :-: | :-: | :-: |
> | $\sqrt{}$ | $\times$ | $\times$ | 60ms | 47.97 |
> | $\sqrt{}$ | $\sqrt{}$ | $\times$ | 55ms (1.09x) |
> | $\sqrt{}$ | $\sqrt{}$ | $\sqrt{}$ | 48ms (1.25x) |

---

> ### Author Response · Authors · 2025-11-22
> **Response to Reviewer HgWG (2/4)**
>
> ## W4: Logit init relies on a good starting mask (Top-N or SparseGPT). The sensitivity to poorer inits and to the magnitude hyperparameter is only partly explored.
>
> We thank the reviewer for pointing out these issues.
>
> **We would like to clarify that random initialization is not ineffective, but rather inefficient to train for RL-based methods.** Unlike gradient-based methods, RL methods typically converge more slowly, so a good initialization can significantly shorten the training process.
>
> To address the reviewer's concerns, we initialized the 7B model with different prior masks for 20000 steps and report the corresponding training results as follows (accuracy is the averaged accuracy on the datasets above):
>
> |     | Random | Top-K | Wanda | GBLM | SparseGPT |
> | :-: | :-: | :-: | :-: | :-: | :-: |
> | LLaMA-7B  | 30.27 | 45.97 | 46.71 | 46.56 | 47.97 |
> | MaskPro   | 36.35 | 48.35 | 49.33 | 49.45 | 49.77 |
> |Improvement| +6.08 | +2.38 | +2.62 | +2.89 | +1.80 |
>
> In practice, when using SparseGPT for initialization, the model converges in roughly 10000 steps. With TopK initialization, extending the training to 20000 steps yields a good result. We can see that random initialization can also train the mask, but the training process is slow.
>
> We further conduct a longer training experiment specifically for random initialization, and the results are as follows:
>
> |     | T=20000 | T=30000 | T=50000 | T=70000 |
> | :-: | :-: | :-: | :-: | :-: |
> | Acc. | 36.35 | 38.43 | 40.74 | 42.37 |
>
> This training process is quite lengthy, and we estimate that completing the full experiment would require at least 300000 steps. Such behavior is consistent with the theoretical convergence rate of RL-based methods, **which is why we do not encourage training from random initialization**. We hope that these two experiments address the reviewer's concerns: it is not that RL-based methods cannot be trained from random initialization, but rather that it is unnecessary, as simple priors such as top-K can significantly shorten the training cycle.
>
> Regarding the role of the other parameter $C$ mentioned by the reviewer, we provide a joint analysis of these two important hyperparameters $\alpha$ and $C$ and their impact on training in our response to Q2.
>
>
> ## W5: The variance-reduction claim depends on a condition. Practical validity across models and tasks is not deeply tested; switching mid-training is suggested but not ablated.
> We thank the reviewer for raising this question. The variance-reduction analysis is intended to provide a theoretical perspective for understanding our method. Since our proof does not yield a strict 'less-than-or-equal-to' inequality, we present an approximate bound in the main text, as the subsequent derivation expresses the expectation using a tracker term. Regarding this assumption, our intention is to highlight the potential theoretical limitations of the analysis, namely that the accelerated training effect of the REINFORCE-baseline method has an upper bound rather than holding indefinitely. We again thank the reviewer for raising this question, and we will add a new remark in the theoretical analysis section to provide a more rigorous explanation.
>
> The stage-wise training strategy is indeed feasible, as it essentially restarts training with the currently obtained mask. Fundamentally, this is a rather trivial engineering approach, while our paper primarily focuses on the modeling principles of MaskPro and its corresponding implementation. We sincerely thank the reviewer for pointing out this issue, and we will revise this suggestion in the next revision.

---

> ### Author Response · Authors · 2025-11-22
> **Response to Reviewer HgWG (3/4)**
>
> ## W6: Only 7B models are reported. No 13B-70B results, where memory and sampling costs and accuracy regressions are more consequential.
>
> We sincerely thank the reviewer for raising this valuable question. During the rebuttal period, we made our best efforts to conduct the following additional experiments.
>
> **(2:4)-Sparsity of LLaMA-13B**
> |       | HellaS. | RACE | PIQA | WinoG. | ARC-E | ARC-C | OBQA | Avg. |
> | :-: | :-: | :-: | :-: | :-: | :-: | :-: | :-: | :-: |
> | Magnitude | **50.10** | 36.84 | 71.76 | 61.88 | 62.29 | 31.74 | 23.40 | 48.29 |
> | SparseGPT | 47.73 | **38.95** | 73.61 | 69.22 | 69.95 | 36.35 | 27.40 | 51.89 |
> | Wanda     | 46.24 | 38.47 | 73.94 | 67.32 | 68.73 | 34.13 | 24.20 | 50.43 |
> | GBLM      | 46.65 | 37.97 | 73.46 | 69.04 | 69.33 | 34.75 | 25.80 | 51.00 |
> | Pruner-0  | 46.15 | 38.85 | 73.13 | 67.24 | 67.52 | 33.89 | 25.20 | 50.28 |
> | MaskPro   | 49.24 | 38.91 | **75.12** | **70.33** | **71.85** | **38.26** | **27.40** | **53.02** |
>
> **(2:4)-Sparsity of LLaMA-30B**
> |       | HellaS. | RACE | PIQA | WinoG. | ARC-E | ARC-C | OBQA | Avg. |
> | :-: | :-: | :-: | :-: | :-: | :-: | :-: | :-: | :-: |
> | Magnitude | 49.57 | 35.69 | 70.24 | 65.59 | 57.32 | 31.66 | 27.80 | 48.27 |
> | SparseGPT | 55.25 | 37.77 | 77.45 | **73.68** | 75.25 | 43.27 | 31.80 | 56.35 |
> | Wanda     | 54.18 | **40.00** | 77.69 | 73.24 | 74.24 | 42.15 | 31.60 | 56.16 |
> | GBLM      | 54.68 | 37.35 | 75.24 | 73.12 | 74.68 | 42.32 | 30.80 | 55.46 |
> | Pruner-0  | 53.69 | 37.13 | 75.86 | 73.04 | 74.23 | 41.25 | 31.20 | 55.20 |
> | MaskPro   | **59.76** | 37.28 | **78.24** | 73.32 | **76.83** | **45.65** | **33.20** | **57.75** |
>
> **(8:16)-Sparsity of LLaMA-7B**
> |       | HellaS. | RACE | PIQA | WinoG. | ARC-E | ARC-C | OBQA | Avg. |
> | :-: | :-: | :-: | :-: | :-: | :-: | :-: | :-: | :-: |
> | Magnitude | 52.27 | 35.02 | 72.74 | 64.48 | 67.68 | 37.03 | 27.20 | 50.92 |
> | SparseGPT | 50.19 | 39.04 | 74.43 | 66.22 | 70.45 | 36.43 | 28.80 | 52.22 |
> | Wanda     | 49.77 | 39.14 | 75.30 | **66.61** | 70.62 | 36.18 | 28.80 | 52.35 |
> | GBLM      | 49.51 | **39.90** | 75.68 | 66.38 | 69.91 | 36.43 | 27.60 | 52.20 |
> | Pruner-0  | 50.12 | 38.68 | 75.22 | 66.13 | 69.93 | 35.48 | 27.80 | 51.91 |
> | MaskPro   | **53.15** | 39.23 | **76.15** | 66.56 | **72.87** | **40.13** | **29.60** | **53.96** |
>
> **(8:16)-Sparsity of LLaMA-13B**
> |       | HellaS. | RACE | PIQA | WinoG. | ARC-E | ARC-C | OBQA | Avg. |
> | :-: | :-: | :-: | :-: | :-: | :-: | :-: | :-: | :-: |
> | Magnitude | 55.43 | 37.51 | 74.48 | 66.06 | 68.94 | 38.05 | 27.60 | 52.58 |
> | SparseGPT | 54.24 | **40.38** | 77.15 | 70.19 | 75.08 | 41.31 | **31.00** | 55.62 |
> | Wanda     | 54.50 | 39.62 | 77.09 | 70.09 | 73.19 | 40.36 | 30.80 | 55.09 |
> | GBLM      | 54.45 | 39.18 | 76.35 | 69.92 | 73.75 | 40.07 | 29.60 | 54.76 |
> | Pruner-0  | 54.11 | 38.64 | 76.28 | 70.41 | 72.92 | 40.55 | 30.00 | 54.70 |
> | MaskPro   | **57.35** | 39.92 | **77.83** | **70.68** | **76.45** | **43.26** | 30.60 | **56.58** |
>
> MaskPro is generally effective across different model scales and sparsity patterns. We hope that the above experiments help alleviate the reviewer's concerns regarding the generality.
>
> ## Q1: How is $p(m|\pi)$ computed efficiently for N-way sampling without replacement? Provide explicit formulas and complexity.
>
> We thank the reviewer for raising this interesting question.
>
> In fact, sampling process only requires providing the probability for each element, after which the selection can be performed using a Bayesian process or a sampling function. In our training procedure, we directly use torch.multinomial and set the config by replacement=False.
>
> **Here we use the 2:4 sparsity setting as an example to illustrate the probability computation process.** The sampling probabilities for the four positions are given as follows $P=[p_1, p_2, p_3, p_4]$. Then
> $$p(m)=(\prod_i p_i^{m_i})(\sum_i\frac{m_i}{1-p_i})=P^m\cdot(\frac{m}{1-P}).sum()$$
>
> These computations are performed elementwise, so they can be obtained directly through matrix Hadamard multiplication (broadcasted multiplication).
>
> **Moreover, regardless of which estimator is used, the linear memory property remains unchanged.** This is because we model (N:M) sparsity as selecting N elements out of M, and using differentiable top-K methods such as Gumbel-TopK only provides an approximation to the sampling process without altering the underlying formulation.
>
> It is worth noting that, besides the standard sampling-without-replacement method, a more efficient exponential clocks method has been proposed in  [1]. This method achieves sub-linear space complexity of $\mathcal{O}(M log(N/M))$ and simultaneously can avoid the need for updating the probabilities of remaining elements at each step of the sampling- without- replacement process, thus saving both computational time and space.
>
> [1] Efraimidis, Pavlos S., and Paul G. Spirakis. "Weighted random sampling with a reservoir." Information processing letters 97.5 (2006): 181-185.

---

> ### Author Response · Authors · 2025-11-22
> **Response to Reviewer HgWG (4/4)**
>
> ## Q2: How sensitive is performance to the initial mask $m_0$ and the logit magnitude $C$?
> We thank the reviewer for raising this meaningful question. We provided exploratory analysis and empirical results regarding initialization in our response to W4. Here, we further discuss the roles of the remaining two key hyperparameters.
>
> In our PG update, the parameter $\alpha$ is used to track a stable estimate of the current baseline and prevent it from being overly influenced by the stochastic variance of sampled losses. Conceptually, this plays the same role as $\beta_1$ or $\beta_2$ in the Adam optimizer. To examine its sensitivity, we conducted the following set of experiments,
>
> | $\alpha$ | 0.7 | 0.9 | 0.95 | 0.99 | 0.995 |
> | :-: | :-: | :-: | :-: | :-: | :-: |
> | LLaMA-7B  | 34.25 | 49.28 | 49.37 | **49.62** | 49.21 |
> | LLaMA-13B | 38.68 | 52.23 | 52.78 | **53.02** | 52.74 |
>
> In our experiments, we found that using $\alpha=0.99$ consistently across all tasks provides the most stable and reliable performance. Therefore, we only report the selection of $0.99$ for reproduction. This hyperparameter requires almost no additional tuning for all models.
>
> And in the initialization, the parameter $C$ is used for stable sampling space. A detailed explanation is provided in Appendix A.2. If $C$ is set too small, a single sampling step has a high probability of producing a poor mask, which can lead to a severe imbalance between positive and negative samples during training, ultimately hindering the learning process of combinatorial optimization. Therefore, choosing a sufficiently large $C$ during initialization allows the training to remain stable. We evaluated different values and the results are as follows:
>
> | $C$ | 8 | 9 | 10 | 11 | 12 |
> | :-: | :-: | :-: | :-: | :-: | :-: |
> | LLaMA-7B  | - | 49.17 | **49.62** | 49.59 | 49.55 |
> | LLaMA-13B | - | 50.45 | 52.94 | **53.02** | 52.99 |
>
> A smaller value of $C$ tends to cause training instability. Once it exceeds a certain threshold, the training becomes highly stable.
>
> ## Q3: Give wall-clock profiles: percent time in sampling, forward passes, and PG updates. Can cached logits or Gumbel-Top-k variants reduce sampling cost further?
> We thank the reviewer for raising this meaningful question. Regarding the time profiles within each training step, we have reported the experimental results in our response to W2. Here, we additionally explore two alternative accelerated sampling strategies along with their corresponding results.
>
> |          | torch.multinomial | Acc. | Naive Gumbel-TopK | Acc. | Gaussion-TopK | Acc.  |
> | :-: | :-: | :-: | :-: | :-: | :-: | :-: |
> | 7B  | 2.328s | 49.62 | 1.821s (1.27x) | 49.24 (-0.38) | 1.496s (1.58x) | 48.84 (-0.78) |
> | 13B | 4.739s | 53.02 | 3.645s (1.30x) | 52.59 (-0.43) | 3.061s (1.55x) | 52.22 (-0.80) |
>
> Within an acceptable error range, the training time can be further reduced. However, we still recommend using higher-precision sampling methods, as the current training time requirement of MaskPro is already quite reasonable. On the impact of randomness on experiments, RL methods rely on sampling, so they are generally less sensitive to random seeds compared with gradient-based methods, and tend to exhibit stronger robustness across settings.
>
> ## Q4: Show end-to-end throughput and latency on A100/H100 with vendor N:M kernels before and after MaskPro, at fixed accuracy. Accuracy-only tables are insufficient to claim deployment readiness.
>
> We provided a simple SparseGEMM-based evaluation using PyTorch's integrated support in our response to W3. As we mentioned above, fully and comprehensively deploying N:M sparsity is a highly involved engineering task, with numerous implementation details, including operator-level optimizations, that extend far beyond the scope of this paper.
>
> We genuinely appreciate the reviewers technical expertise and fully acknowledge the importance of deployment readiness. At the same time, we hope the reviewer can also take into account the core contribution of this paper.
>
> Our core contribution is to provide a theoretically grounded optimization framework for selecting effective N:M sparsity patterns, which is fundamentally orthogonal to the engineering aspects of deployment.

---

> ### Author Response · Authors · 2025-11-27
> **Invitation to rolling discussion for the possible remaining concerns**
>
> Dear Reviewer HgWG,
>
> **Thank you once again for the time and effort you dedicated to reviewing our submission, as well as for your positive assessment of our work.** In this rebuttal, we have incorporated additional clarifications, explanations, experiments, and discussions to address your concerns, and we have followed your suggestions to further improve our submission.
>
> As we are approaching the end of the ICLR public discussion period, would you mind checking our rebuttal and confirming if they have addressed your concerns? We truly appreciate this opportunity to improve our work and shall be most grateful for any feedback you could give to us.

---

> ### Comment · Reviewer_HgWG · 2025-11-28
> **Raising score**
>
> Thanks for the detailed rebuttal! I keep my rating to maintain the acceptance of this paper.

---

> > ### Author Response · Authors · 2025-11-28
> > **Thank you again for your review**
> >
> > Thank you very much for your response and the positive feedback of our work. Although the final rate cannot be updated due to system constraints, we truly appreciate your willingness to raise it.
> >
> > Most of the revision in our submission are made based on the valuable suggestions from you, which has indeed made the current version much more complete compared to our initial submission. The experiments on hyperparameters and the time-based validation are indeed very important supplementary studies, and we would like to express our gratitude once again. We will further refine our paper according to the points discussed with you.

---

### Official Review · Reviewer_rhLs · 2025-10-30

**Soundness:** 1
**Presentation:** 2
**Contribution:** 2
**Rating:** 2
**Confidence:** 3

**Summary:**

This paper introduces MaskPro, a memory-efficient probabilistic framework for learning strict (N:M)-structured sparsity in large language models (LLMs). The method models each group of M weights using a categorical distribution and generates sparse masks via N-way sampling without replacement, reducing the memory complexity from exponential to linear. It also proposes a new policy gradient estimator (PGE) that replaces the vanilla loss metric with loss residuals and employs a smoothing tracker to stabilize optimization. Experiments on several 7B-scale models show moderate improvements in performance and significant reductions in memory usage compared to prior approaches such as MaskLLM.

**Strengths:**

- The proposed update rule based on loss residuals, combined with a moving average tracker, is an interesting and practical way to stabilize the inherently noisy gradients in policy-gradient-based mask learning. The authors demonstrate theoretically that this estimator remains unbiased while reducing variance.
- MaskPro achieves substantial memory and computational efficiency, requiring roughly 36 GB of memory compared to over 300 GB for MaskLLM, while running on a single GPU.
- The method performs well even when trained with very small datasets (sometimes as few as one sample), showcasing robustness to data scarcity.

**Weaknesses:**

- The experimental evaluation focuses solely on mid-sized (7B) models and only on (2:4) sparsity configurations, with minimal exploration of other ratios or architectures. There is no validation on larger or smaller models, no finetuning experiments, and no runtime or latency measurements to verify practical benefits. As a result, the empirical evidence supporting MaskPro’s claimed generality and scalability remains limited.
- The method's success is critically dependent on a complex initialization strategy. Standard random or zero initializations are "ineffective", and the training fails if the initial logits magnitude $C$ is not large enough.
- The comparisons with baselines such as MaskLLM, SparseGPT, and Wanda are not entirely fair, as these methods differ in their reliance on fine-tuning versus frozen weights, training data sizes, and initialization strategies. Moreover, MaskPro benefits from initialization using precomputed SparseGPT masks, which gives it an advantage that is not acknowledged or ablated. The paper would be stronger if these factors were controlled more carefully.
- Although the authors emphasize MaskPro’s efficiency, the claimed linear-space scaling overlooks constant factors associated with sampling and softmax operations, and the actual runtime improvements are not measured. The claim that the method can train effectively with one sample seems implausible without relying on strong priors, undermining the claim of data robustness. More empirical validation is needed to substantiate these statements.
- The algorithmic description omits critical implementation details such as how N-way sampling without replacement is realized efficiently, how randomness affects reproducibility, and how logits are updated in parallel. The notation (e.g., the $\oplus$ operator) is nonstandard and lacks intuition, while figures are schematic and do not convey architectural structure or ablation results. Overall, the presentation is mathematically heavy and could be made clearer.

**Questions:**

- Can the authors provide wall-clock runtime and throughput comparisons with MaskLLM and rule-based baselines?
- How sensitive is MaskPro to the smoothing parameter $α$ and the initialization constant $C$?
- What happens when masks are initialized randomly instead of using SparseGPT priors?

---

> ### Author Response · Authors · 2025-11-22
> **Response to Reviewer rhLs (1/6)**
>
> ## W1: The experimental evaluation focuses solely on mid-sized (7B) models and only on (2:4) sparsity configurations, with minimal exploration of other ratios or architectures. There is no validation on larger or smaller models, no finetuning experiments, and no runtime or latency measurements to verify practical benefits. As a result, the empirical evidence supporting MaskPro's claimed generality and scalability remains limited.
> We sincerely thank the reviewer for raising this valuable question. In the original submission, we mainly evaluated the 7B model under the (2:4) and (4:8) sparsity ratios. During the rebuttal period, we made our best efforts to conduct the following additional experiments to further validate the effectiveness of MaskPro:
>
> **(2:4)-Sparsity of LLaMA-13B**
> |       | HellaS. | RACE | PIQA | WinoG. | ARC-E | ARC-C | OBQA | Avg. |
> | :-: | :-: | :-: | :-: | :-: | :-: | :-: | :-: | :-: |
> | Magnitude | **50.10** | 36.84 | 71.76 | 61.88 | 62.29 | 31.74 | 23.40 | 48.29 |
> | SparseGPT | 47.73 | **38.95** | 73.61 | 69.22 | 69.95 | 36.35 | 27.40 | 51.89 |
> | Wanda     | 46.24 | 38.47 | 73.94 | 67.32 | 68.73 | 34.13 | 24.20 | 50.43 |
> | GBLM      | 46.65 | 37.97 | 73.46 | 69.04 | 69.33 | 34.75 | 25.80 | 51.00 |
> | Pruner-0  | 46.15 | 38.85 | 73.13 | 67.24 | 67.52 | 33.89 | 25.20 | 50.28 |
> | MaskPro   | 49.24 | 38.91 | **75.12** | **70.33** | **71.85** | **38.26** | **27.40** | **53.02** |
>
> **(2:4)-Sparsity of LLaMA-30B**
> |       | HellaS. | RACE | PIQA | WinoG. | ARC-E | ARC-C | OBQA | Avg. |
> | :-: | :-: | :-: | :-: | :-: | :-: | :-: | :-: | :-: |
> | Magnitude | 49.57 | 35.69 | 70.24 | 65.59 | 57.32 | 31.66 | 27.80 | 48.27 |
> | SparseGPT | 55.25 | 37.77 | 77.45 | **73.68** | 75.25 | 43.27 | 31.80 | 56.35 |
> | Wanda     | 54.18 | **40.00** | 77.69 | 73.24 | 74.24 | 42.15 | 31.60 | 56.16 |
> | GBLM      | 54.68 | 37.35 | 75.24 | 73.12 | 74.68 | 42.32 | 30.80 | 55.46 |
> | Pruner-0  | 53.69 | 37.13 | 75.86 | 73.04 | 74.23 | 41.25 | 31.20 | 55.20 |
> | MaskPro   | **59.76** | 37.28 | **78.24** | 73.32 | **76.83** | **45.65** | **33.20** | **57.75** |
>
> **(8:16)-Sparsity of LLaMA-7B**
> |       | HellaS. | RACE | PIQA | WinoG. | ARC-E | ARC-C | OBQA | Avg. |
> | :-: | :-: | :-: | :-: | :-: | :-: | :-: | :-: | :-: |
> | Magnitude | 52.27 | 35.02 | 72.74 | 64.48 | 67.68 | 37.03 | 27.20 | 50.92 |
> | SparseGPT | 50.19 | 39.04 | 74.43 | 66.22 | 70.45 | 36.43 | 28.80 | 52.22 |
> | Wanda     | 49.77 | 39.14 | 75.30 | **66.61** | 70.62 | 36.18 | 28.80 | 52.35 |
> | GBLM      | 49.51 | **39.90** | 75.68 | 66.38 | 69.91 | 36.43 | 27.60 | 52.20 |
> | Pruner-0  | 50.12 | 38.68 | 75.22 | 66.13 | 69.93 | 35.48 | 27.80 | 51.91 |
> | MaskPro   | **53.15** | 39.23 | **76.15** | 66.56 | **72.87** | **40.13** | **29.60** | **53.96** |
>
> **(8:16)-Sparsity of LLaMA-13B**
> |       | HellaS. | RACE | PIQA | WinoG. | ARC-E | ARC-C | OBQA | Avg. |
> | :-: | :-: | :-: | :-: | :-: | :-: | :-: | :-: | :-: |
> | Magnitude | 55.43 | 37.51 | 74.48 | 66.06 | 68.94 | 38.05 | 27.60 | 52.58 |
> | SparseGPT | 54.24 | **40.38** | 77.15 | 70.19 | 75.08 | 41.31 | **31.00** | 55.62 |
> | Wanda     | 54.50 | 39.62 | 77.09 | 70.09 | 73.19 | 40.36 | 30.80 | 55.09 |
> | GBLM      | 54.45 | 39.18 | 76.35 | 69.92 | 73.75 | 40.07 | 29.60 | 54.76 |
> | Pruner-0  | 54.11 | 38.64 | 76.28 | 70.41 | 72.92 | 40.55 | 30.00 | 54.70 |
> | MaskPro   | **57.35** | 39.92 | **77.83** | **70.68** | **76.45** | **43.26** | 30.60 | **56.58** |
>
> MaskPro is generally effective across different model scales and sparsity patterns. We hope that the above experiments help alleviate the reviewer's concerns regarding the generality of our method. If the reviewers expect additional experiments, we will make every effort to further validate them.
>
> Regarding the reviewer's questions about training resources and runtime, we note that these points also appear in a later comment. To avoid redundancy and ensure clarity, we provide a dedicated response at the corresponding question.

---

> ### Author Response · Authors · 2025-11-22
> **Response to Reviewer rhLs (2/6)**
>
> ## W2: The method's success is critically dependent on a complex initialization strategy. Standard random or zero initializations are "ineffective", and the training fails if the initial logits magnitude $C$ is not large enough.
> We thank the reviewer for pointing out these issues.
>
> **We would like to clarify that random initialization is not ineffective, but rather inefficient to train for RL-based methods.** Unlike gradient-based methods, RL methods typically converge more slowly, so a good initialization can significantly shorten the training process.
>
> To address the reviewer's concerns, we initialized the 7B model with different prior masks for 20000 steps and report the corresponding training results as follows (accuracy is the averaged accuracy on the datasets above):
>
> |     | Random | Top-K | Wanda | GBLM | SparseGPT |
> | :-: | :-: | :-: | :-: | :-: | :-: |
> | LLaMA-7B  | 30.27 | 45.97 | 46.71 | 46.56 | 47.97 |
> | MaskPro   | 36.35 | 48.35 | 49.33 | 49.45 | 49.77 |
> |Improvement| +6.08 | +2.38 | +2.62 | +2.89 | +1.80 |
>
> In practice, when using SparseGPT for initialization, the model converges in roughly 10000 steps. With TopK initialization, extending the training to 20000 steps yields a good result. We can see that random initialization can also train the mask, but the training process is slow.
>
> We further conduct a longer training experiment specifically for random initialization, and the results are as follows:
>
> |     | T=20000 | T=30000 | T=50000 | T=70000 |
> | :-: | :-: | :-: | :-: | :-: |
> | Acc. | 36.35 | 38.43 | 40.74 | 42.37 |
>
> This training process is quite lengthy, and we estimate that completing the full experiment would require at least 300000 steps. Such behavior is consistent with the theoretical convergence rate of RL-based methods, **which is why we do not encourage training from random initialization**. We hope that these two experiments address the reviewer's concerns: it is not that RL-based methods cannot be trained from random initialization, but rather that it is unnecessary, as simple priors such as top-K can significantly shorten the training cycle.
>
> Regarding the role of the other parameter $C$ mentioned by the reviewer, we provide a joint analysis of these two important hyperparameters $\alpha$ and $C$ and their impact on training in our response to Q2.

---

> ### Author Response · Authors · 2025-11-22
> **Response to Reviewer rhLs (3/6)**
>
> ## W3: The comparisons with baselines such as MaskLLM, SparseGPT, and Wanda are not entirely fair, as these methods differ in their reliance on fine-tuning versus frozen weights, training data sizes, and initialization strategies. Moreover, MaskPro benefits from initialization using precomputed SparseGPT masks, which gives it an advantage that is not acknowledged or ablated. The paper would be stronger if these factors were controlled more carefully.
> We thank the reviewer for raising this question. To clarify the fairness of the experimental setup, we provide the following table for detailed comparison.
>
> First, we follow the experimental setup of MaskLLM, where all methods train only the mask without any parameter finetuning. Simultaneously optimizing both the weights and the mask is inherently an NP-hard problem. Current approaches commonly used in the community rely on approximations: either **treating the (N:M) sparsity as a soft constraint** during optimization, or empirically finetuning the parameters corresponding to the learned mask on a calibration dataset to obtain a performance gain. In contrast, all the baselines in our paper treat it as a **hard constraint**.
>
> The comparison of training sample sizes is quite straightforward. In fact, MaskLLM is a gradient-based training method, and thus it indeed requires a very large dataset, containing approximately 512k samples. However, our method relies on policy-gradient estimation, where the core requirement is not the number of samples but the precision of the loss evaluation (reward) for each sample. In practice, we only need a few hundred data samples to perform effective training.
>
> Rule-based methods do not require initialization, while training-based methods do. As noted above, initialization mainly helps speed up training. We do not encourage random initialization because RL methods converge more slowly than gradient-based ones. In practice, a simple prior such as top-K initialization can greatly reduce the required training steps.
>
> In summary, in response to the reviewer's concerns about the experimental background, we provide the following table:
>
> |       | Type | f.t. weight | Dataset Size | Initialization |
> | :-: | :-: | :-: | :-: | :-: |
> | SparseGPT | Rule-based | $\times$ | 128 | - |
> | Wanda     | Rule-based | $\times$ | 128 | - |
> | GBLM      | Rule-based | $\times$ | 128 | - |
> | Pruner-0  | Rule-based | $\times$ | 128 | - |
> | MaskLLM   | Learning-based | $\times$ | about 520k | SparseGPT / Random |
> | MaskPro   | Learning-based | $\times$ | 128 | SparseGPT / TopK |
>
> To the best of our knowledge, MaskLLM is the first method to apply (N:M) sparsity as a hard constraint in the optimization of LLMs, which is an impressive contribution. However, its training cost is extremely high. Therefore, our core contribution is to formulate the (N:M) sparsity as a hard-constraint optimization problem and train it using linear probability modeling together with policy-gradient updates. This allows us to complete the training process under the same resource requirements as rule-based methods.

---

> ### Author Response · Authors · 2025-11-22
> **Response to Reviewer rhLs (4/6)**
>
> ## W4: Although the authors emphasize MaskPro's efficiency, the claimed linear-space scaling overlooks constant factors associated with sampling and softmax operations, and the actual runtime improvements are not measured. The claim that the method can train effectively with one sample seems implausible without relying on strong priors, undermining the claim of data robustness. More empirical validation is needed to substantiate these statements.
> We thank the reviewer for raising this valuable questions.
>
> It is worth noting that, besides the standard sampling-without-replacement method, a more efficient exponential clocks method has been proposed in [1]. This method achieves sub-linear space complexity of $\mathcal{O}(M log(N/M))$ and simultaneously can avoid the need for updating the probabilities of remaining elements at each step of the sampling- without- replacement process, thus saving both computational time and space.
>
> The term linear space refers to the memory required for storing logits. The reviewer's question conflates the concepts of time and memory. Our formulation is designed to address the memory overhead in MaskLLM, which directly samples from the original combinatorial space and therefore incurs substantial memory cost. By introducing a joint sampling procedure, we ensure that the memory requirement for logits scales linearly with the size of the original parameter space, which is what we refer to as linear space. As for the sampling time, we provide the single-step sampling time for the 7B and 13B models under different coefficient patterns:
>
> |          | (2:4) | (4:8) | (8:16) |
> | :-: | :-: | :-: | :-: |
> | LLaMA-7B  | 2.328s | 1.334s | 0.794s |
> | LLaMA-13B | 4.739s | 2.692s | 1.574s |
>
> We can observe that doubling the model size roughly doubles the sampling time. Another interesting observation is that the sampling time of (N:M)-Sparsity depends on $M$. With the same model size, a larger $M$ leads to shorter sampling time, due to parallel optimizations in the sampling process. For a $d$-dimensional model, there are a total of $\frac{d}{M}$ sampling groups. Although increasing $N$ and $M$ makes each group more expensive to sample, the total number of parallel groups decreases proportionally. This reduction in the number of groups results in a more favorable computation pattern for hardware. Consequently, for more complex (N:M)-Sparsity, the time required for a single sampling step can actually be lower.
>
> Regarding the second question, we conducted training with different initialization strategies using only one sample, and the results remained stable.
>
> |     | Random | Top-K | Wanda | GBLM | SparseGPT |
> | :-: | :-: | :-: | :-: | :-: | :-: |
> | with 128 data samples  | 36.35 | 49.35 | 49.33 | 49.45 | 49.77 |
> | with 1 data sample     | 35.97 | 49.12 | 49.04 | 49.21 | 49.18 |
>
> We would like to clarify that MaskPro is indeed not very sensitive to the number of samples. The essence of RL-based methods lies in accurately estimating and constructing the reward, rather than relying on large data volumes. While we do not deny that using a larger dataset may yield further improvements, the performance obtained with only a few hundred samples is already very close. We hope that this experiment alleviates the reviewer's concerns.
>
> [1] Efraimidis, Pavlos S., and Paul G. Spirakis. "Weighted random sampling with a reservoir." Information processing letters 97.5 (2006): 181-185.

---

> ### Author Response · Authors · 2025-11-22
> **Response to Reviewer rhLs (5/6)**
>
> ## W5: The algorithmic description omits critical implementation details such as how N-way sampling without replacement is realized efficiently, how randomness affects reproducibility, and how logits are updated in parallel. The notation (e.g., the $\oplus$ operator) is nonstandard and lacks intuition, while figures are schematic and do not convey architectural structure or ablation results. Overall, the presentation is mathematically heavy and could be made clearer.
>
> We sincerely thank the reviewer for raising these questions. We address them one by one below.
>
> We use torch.multinomial function for (N:M)-sparsity sampling, which simulates the sampling process through a lookup-based mechanism and provides high accuracy. The forward pass is implemented through a standard wrapper function. Specifically, we wrap the linear layer with an additional mask parameter and integrate the mask computation directly inside the linear operation. This design avoids modifying PyTorch's computation graph and enables efficient inference. The logits updates are computed entirely through matrix calculation, and PyTorch's built-in libraries already provide the necessary parallelization. To further illustrate the implementation details, we report the per-step training time as follows:
>
> |          | Mask Sampling | Ratio | Forward | Ratio | PG Update | Ratio |
> | :-: | :-: | :-: | :-: | :-: | :-: | :-: |
> | 7B  | 2.328s | 85.94% | 0.062s | 2.29% | 0.319s | 11.77% |
> | 13B | 4.739s | 85.83% | 0.139s | 2.52% | 0.644s | 11.65% |
>
> The main source of time consumption comes from the sampling process, which is also a common phenomenon in the RL community. Following the suggestion of reviewer HgWG, we explored several alternative sampling strategies during the rebuttal period and report the results below:
>
> |          | torch.multinomial | Acc. | Naive Gumbel-TopK | Acc. | Gaussion-TopK | Acc.  |
> | :-: | :-: | :-: | :-: | :-: | :-: | :-: |
> | 7B  | 2.328s | 49.62 | 1.821s (1.27x) | 49.24 (-0.38) | 1.496s (1.58x) | 48.84 (-0.78) |
> | 13B | 4.739s | 53.02 | 3.645s (1.30x) | 52.59 (-0.43) | 3.061s (1.55x) | 52.22 (-0.80) |
>
> Within an acceptable error range, the training time can be further reduced. However, we still recommend using higher-precision sampling methods, as the current training time requirement of MaskPro is already quite reasonable. On the impact of randomness on experiments, RL methods rely on sampling, so they are generally less sensitive to random seeds compared with gradient-based methods, and tend to exhibit stronger robustness across settings.
>
> Regarding the notation, we follow the common conventions in combinatorial optimization and information theory. We apologize for the mathematical complexity that such notation may introduce. However, the purpose of this work is to theoretically establish how a linear probability space can be formulated under strict N:M sparsity constraints. One of our original motivations for writing this paper is to promote this modeling process so that N:M sparse learning goes beyond engineering-level approximations and is instead grounded in a well-defined optimization problem.

---

> ### Author Response · Authors · 2025-11-22
> **Response to Reviewer rhLs (6/6)**
>
> ## Q1: Can the authors provide wall-clock runtime and throughput comparisons with MaskLLM and rule-based baselines?
> We thank the reviewer for raising this question, and we summarize the time and memory of processing the 7B model on H100 devices as follows.
>
> |          | Magnitude | Wanda | SparseGPT | GBLM | Pruner-0 | MaskLLM | MaskPro |
> | :-: | :-: | :-: | :-: | :-: | :-: | :-: | :-: |
> | Type  | Rule-based | Rule-based | Rule-based | Rule-based | Rule-based | **Learning-based** | **Learning-based** |
> | Time  | less than 1 min | 1 min | 3min | 0.7h | 0.8h | 1600h | 7h |
> | Memory  | 12.82G | 22.20G | 21.25G | 26.87G | 26.87G | 331.16G | 35.90G |
>
> We do not deny that learning-based methods require more time than rule-based approaches, which is indeed expected. However, learning-based methods also achieve higher performance, so a direct comparison purely based on training time should be interpreted with caution. **Our main objective is to substantially reduce the training cost of the learning-based methods for the strict (N:M)-Sparsity on the limited hardware budgets.**
>
>
> ## Q2: How sensitive is MaskPro to the smoothing parameter $\alpha$ and the initialization constant $C$?
> We thank the reviewer for raising this meaningful question.
>
> In our PG update, the parameter $\alpha$ is used to track a stable estimate of the current baseline and prevent it from being overly influenced by the stochastic variance of sampled losses. Conceptually, this plays the same role as $\beta_1$ or $\beta_2$ in the Adam optimizer. To examine its sensitivity, we conducted the following set of experiments,
>
> | $\alpha$ | 0.7 | 0.9 | 0.95 | 0.99 | 0.995 |
> | :-: | :-: | :-: | :-: | :-: | :-: |
> | LLaMA-7B  | 34.25 | 49.28 | 49.37 | **49.62** | 49.21 |
> | LLaMA-13B | 38.68 | 52.23 | 52.78 | **53.02** | 52.74 |
>
> In our experiments, we found that using $\alpha=0.99$ consistently across all tasks provides the most stable and reliable performance. Therefore, we only report the selection of $0.99$ for reproduction. This hyperparameter requires almost no additional tuning for all models.
>
> And in the initialization, the parameter $C$ is used for stable sampling space. A detailed explanation is provided in Appendix A.2. If $C$ is set too small, a single sampling step has a high probability of producing a poor mask, which can lead to a severe imbalance between positive and negative samples during training, ultimately hindering the learning process of combinatorial optimization. Therefore, choosing a sufficiently large $C$ during initialization allows the training to remain stable. We evaluated different values and the results are as follows:
>
> | $C$ | 8 | 9 | 10 | 11 | 12 |
> | :-: | :-: | :-: | :-: | :-: | :-: |
> | LLaMA-7B  | - | 49.17 | **49.62** | 49.59 | 49.55 |
> | LLaMA-13B | - | 50.45 | 52.94 | **53.02** | 52.99 |
>
> A smaller value of $C$ tends to cause training instability. Once it exceeds a certain threshold, the training becomes highly stable.
>
> ## Q3: What happens when masks are initialized randomly instead of using SparseGPT priors?
> See answers of W2.

---

> ### Author Response · Authors · 2025-11-27
> **Invitation to rolling discussion for the possible remaining concerns**
>
> Dear Reviewer rhLs,
>
> **Thank you once again for the time and effort you dedicated to reviewing our submission, as well as for your valuable suggestions of our work.** In this rebuttal, we have incorporated additional clarifications, explanations, experiments, and discussions to address your concerns, and we have followed your suggestions to further improve our submission.
>
> As we are approaching the end of the ICLR public discussion period, would you mind checking our rebuttal and confirming if they have addressed your concerns? We truly appreciate this opportunity to improve our work and shall be most grateful for the feedback.

---

### Official Review · Reviewer_4Q5R · 2025-10-30

**Soundness:** 3
**Presentation:** 4
**Contribution:** 3
**Rating:** 8
**Confidence:** 5

**Summary:**

This paper presents MaskPro, a linear-space probabilistic framework for learning strict N:M sparsity in LLMs. The work addresses the limitations of existing approaches, rule-based methods that introduce bias and learning-based ones that demand prohibitive computational cost, by proposing a lightweight probabilistic formulation. It models each group of weights with a simple categorical distribution and learns sparsity patterns through a refined policy gradient update that incorporates loss residuals and a moving average tracker, improving both stability and convergence. Empirically, MaskPro achieves strong results across multiple 7B-scale models, reaching good accuracy while reducing memory usage by an order of magnitude and requiring only a handful of training samples. The approach is conceptually clear, theoretically supported, and practically appealing.

**Strengths:**

1. The reformulation of the (N:M) sparsity problem into a linear-space probabilistic model is both interesting and convincing. It reduces the memory requirement from exponential to linear scale and lowers the data demand compared with prior learning-based methods.
2. MaskPro demonstrates consistently strong performance across diverse benchmarks and model backbones, achieving comparable or superior accuracy to established baselines while maintaining high efficiency.
3. The approach is remarkably data-efficient, as evidenced in Figure 3, where stable results are obtained even with minimal training samples, making the method highly practical for large-scale applications.

**Weaknesses:**

1. While the results on 2:4 sparsity are strong, it remains to be seen how well MaskPro generalizes to other configurations, such as 8:16. These settings involve significantly larger combinatorial spaces, roughly 12,870 combinations per group in MaskLLM, and would serve as a valuable test of the proposed probabilistic formulation’s scalability.
2. It would be interesting to explore whether MaskPro can be extended to jointly optimize both the sparsity pattern and the LLM parameters. Such a joint optimization could potentially yield further improvements in performance and adaptability.

**Questions:**

Please see the weaknesses.

---

> ### Author Response · Authors · 2025-11-22
> **Response to Reviewer 4Q5R (1/2)**
>
> ## W1: While the results on 2:4 sparsity are strong, it remains to be seen how well MaskPro generalizes to other configurations, such as 8:16. These settings involve significantly larger combinatorial spaces, roughly 12,870 combinations per group in MaskLLM, and would serve as a valuable test of the proposed probabilistic formulation's scalability.
>
> We sincerely thank the reviewer for raising this valuable question.
>
> We have made every effort during the rebuttal period to conduct the following experiments to validate the scalability of our model. A larger (N:M)-Sparsity indeed introduces a much larger sampling space, which undoubtedly increases the training difficulty. To evaluate its trainability, we trained the (8:16)-Sparsity with twice the number of iterations used for the (2:4)-Sparse model, and obtained the following results:
>
> **(8:16)-Sparsity of LLaMA-7B**
> |       | HellaS. | RACE | PIQA | WinoG. | ARC-E | ARC-C | OBQA | Avg. |
> | :-: | :-: | :-: | :-: | :-: | :-: | :-: | :-: | :-: |
> | Magnitude | 52.27 | 35.02 | 72.74 | 64.48 | 67.68 | 37.03 | 27.20 | 50.92 |
> | SparseGPT | 50.19 | 39.04 | 74.43 | 66.22 | 70.45 | 36.43 | 28.80 | 52.22 |
> | Wanda     | 49.77 | 39.14 | 75.30 | **66.61** | 70.62 | 36.18 | 28.80 | 52.35 |
> | GBLM      | 49.51 | **39.90** | 75.68 | 66.38 | 69.91 | 36.43 | 27.60 | 52.20 |
> | Pruner-0  | 50.12 | 38.68 | 75.22 | 66.13 | 69.93 | 35.48 | 27.80 | 51.91 |
> | MaskPro   | **53.15** | 39.23 | **76.15** | 66.56 | **72.87** | **40.13** | **29.60** | **53.96** |
>
> **(8:16)-Sparsity of LLaMA-13B**
> |       | HellaS. | RACE | PIQA | WinoG. | ARC-E | ARC-C | OBQA | Avg. |
> | :-: | :-: | :-: | :-: | :-: | :-: | :-: | :-: | :-: |
> | Magnitude | 55.43 | 37.51 | 74.48 | 66.06 | 68.94 | 38.05 | 27.60 | 52.58 |
> | SparseGPT | 54.24 | **40.38** | 77.15 | 70.19 | 75.08 | 41.31 | **31.00** | 55.62 |
> | Wanda     | 54.50 | 39.62 | 77.09 | 70.09 | 73.19 | 40.36 | 30.80 | 55.09 |
> | GBLM      | 54.45 | 39.18 | 76.35 | 69.92 | 73.75 | 40.07 | 29.60 | 54.76 |
> | Pruner-0  | 54.11 | 38.64 | 76.28 | 70.41 | 72.92 | 40.55 | 30.00 | 54.70 |
> | MaskPro   | **57.35** | 39.92 | **77.83** | **70.68** | **76.45** | **43.26** | 30.60 | **56.58** |
>
> Under this sparsity pattern, the memory requirement of MaskLLM becomes extremely large, even exceeding the resource demands commonly used in the community to train models with hundreds of billions of parameters. Moreover, our MaskPro approach introduce minor training cost, while achieving better results than rule-based methods.
>
> Additionally, we would like to report another interesting observation. We further summarize the single-step sampling time for the 7B and 13B models under different coefficient patterns:
>
> |          | (2:4) | (4:8) | (8:16) |
> | :-: | :-: | :-: | :-: |
> | LLaMA-7B  | 2.328s | 1.334s | 0.794s |
> | LLaMA-13B | 4.739s | 2.692s | 1.574s |
>
> We use the torch.multinomial to sample masks within each group. We can observe that doubling the model size roughly doubles the sampling time. Another interesting observation is that the sampling time of (N:M)-Sparsity depends on $M$. With the same model size, a larger $M$ leads to shorter sampling time, due to parallel optimizations in the sampling process. For a $d$-dimensional model, there are a total of $\frac{d}{M}$ sampling groups. Although increasing $N$ and $M$ makes each group more expensive to sample, the total number of parallel groups decreases proportionally. This reduction in the number of groups results in a more favorable computation pattern for hardware. Consequently, for more complex (N:M)-Sparsity, the time required for a single sampling step can actually be lower. **Large M is a GPU-friendly selection.**
>
> We hope that these additional experiments further address the reviewer's concerns. If the reviewers are interested in any other experimental settings, we would be more than happy to explore them as well.

---

> ### Author Response · Authors · 2025-11-22
> **Response to Reviewer 4Q5R (2/2)**
>
> ## W2: It would be interesting to explore whether MaskPro can be extended to jointly optimize both the sparsity pattern and the LLM parameters. Such a joint optimization could potentially yield further improvements in performance and adaptability.
> We sincerely thank the reviewer for raising this highly valuable academic question!
>
> Solving this problem is indeed one of the directions we are actively exploring. Existing research has shown that jointly optimizing the mask and the weights is an NP-hard problem [1,2], making it highly challenging to find a stationary point from an optimization perspective.
>
> Among existing approaches for addressing this problem, a classical idea is to convert the (N:M) constraint into a soft constraint and jointly optimize it using a corresponding regularization loss. This strategy has the advantage of enabling end-to-end training. However, its drawbacks are also evident: soft constraints are often not well satisfied, and even after training, an additional approximation step on a calibration set is needed to obtain a valid mask. This post-processing step typically leads to a significant drop in performance. Another major line of approaches relies on stage-wise training, where the mask and the weights are optimized alternately. Such methods are more like engineering approximations rather than true optimizations that approach the optimal solution. To date, jointly finding the optimal mask and weights remains an open challenge.
>
> We have recently been exploring a dual-optimization formulation to progressively approach a joint solution to this problem. Preliminary results suggest that such a formulation may help decouple the combinatorial structure of the mask from the continuous optimization of the weights, potentially enabling a more stable and theoretically grounded training process. While this direction is still in its early stages, we believe it offers a promising pathway toward overcoming the inherent NP-hardness and bringing us closer to a principled solution for simultaneous mask-weight optimization.
>
> We thank again the reviewer for raising this highly professional and academically insightful question. We will include an additional discussion of this research direction in the final section. If you have any additional questions, please feel free to let us know, and we will address them.
>
> [1] Natarajan, Balas Kausik. "Sparse approximate solutions to linear systems." SIAM journal on computing 24.2 (1995): 227-234.
>
> [2] Feige, Uriel. "A threshold of ln n for approximating set cover." Journal of the ACM (JACM) 45.4 (1998): 634-652.

---

> ### Author Response · Authors · 2025-11-27
> **Invitation to rolling discussion for the possible remaining concerns**
>
> Dear Reviewer 4Q5R,
>
> **Thank you once again for the time and effort you dedicated to reviewing our submission, as well as for your positive assessment of our work.** In this rebuttal, we have incorporated additional clarifications, explanations, experiments, and discussions to address your concerns, and we have followed your suggestions to further improve our submission.
>
> As we are approaching the end of the ICLR public discussion period, would you mind checking our rebuttal and confirming if they have addressed your concerns? We truly appreciate this opportunity to improve our work and shall be most grateful for any feedback you could give to us.

---

### Official Review · Reviewer_NXWQ · 2025-11-12

**Soundness:** 4
**Presentation:** 4
**Contribution:** 4
**Rating:** 6
**Confidence:** 3

**Summary:**

The paper introduces a new probabilistic framework for learning hardware-friendly semi-structured sparsity in LLMs. Unlike prior methods, MaskPro reduces both memory and computation overhead by modeling sparsity as an N-way sampling without replacement from categorical distributions over M consecutive weights. It leverages a linear-space parameterization that reduces memory complexity to linear. and optimizes via a refined PGE that replaces the raw loss with a loss residual tracked by a moving average, improving stability and variance reduction. Extensive experiments on multiple 7B LLMs show that MaskPro outperforms baselines while approaching the their accuracy at 10x less memory and training cost.

**Strengths:**

Sound theoretical explanation, linear memory efficiency, and comprehensive validation experiments.

**Weaknesses:**

Experiments focus mainly on 2:4 sparsity and 7B models; scaling to higher sparsity ratios or larger models, e.g., 70B, is not demonstrated. (might due to hardware constraints)

**Questions:**

How sensitive is MaskPro's performance to the choice of $\alpha$?

Does the sampling-without-replacement process still scale linearly in memory and remain computationally tractable for large M? This needs further discussion.

---

> ### Author Response · Authors · 2025-11-22
> **Response to Reviewer NXWQ (1/2)**
>
> ## W1: Experiments focus mainly on 2:4 sparsity and 7B models; scaling to higher sparsity ratios or larger models, e.g., 70B, is not demonstrated. (might due to hardware constraints)
>
> Thank you very much for your suggestion. In the vanilla submission, we also report the results on (4:8)-sparsity (Section A.5 in Appendix). The training resources required for a 70B model are simply too large, and we are currently unable to afford them. To alleviate the reviewer's concerns, we have made our best efforts to include additional experiments with larger model scales and different sparsity patterns to further validate the effectiveness of the proposed method.
>
> **(2:4)-Sparsity of LLaMA-13B**
> |       | HellaS. | RACE | PIQA | WinoG. | ARC-E | ARC-C | OBQA | Avg. |
> | :-: | :-: | :-: | :-: | :-: | :-: | :-: | :-: | :-: |
> | Magnitude | **50.10** | 36.84 | 71.76 | 61.88 | 62.29 | 31.74 | 23.40 | 48.29 |
> | SparseGPT | 47.73 | **38.95** | 73.61 | 69.22 | 69.95 | 36.35 | 27.40 | 51.89 |
> | Wanda     | 46.24 | 38.47 | 73.94 | 67.32 | 68.73 | 34.13 | 24.20 | 50.43 |
> | GBLM      | 46.65 | 37.97 | 73.46 | 69.04 | 69.33 | 34.75 | 25.80 | 51.00 |
> | Pruner-0  | 46.15 | 38.85 | 73.13 | 67.24 | 67.52 | 33.89 | 25.20 | 50.28 |
> | MaskPro   | 49.24 | 38.91 | **75.12** | **70.33** | **71.85** | **38.26** | **27.40** | **53.02** |
>
> **(2:4)-Sparsity of LLaMA-30B**
> |       | HellaS. | RACE | PIQA | WinoG. | ARC-E | ARC-C | OBQA | Avg. |
> | :-: | :-: | :-: | :-: | :-: | :-: | :-: | :-: | :-: |
> | Magnitude | 49.57 | 35.69 | 70.24 | 65.59 | 57.32 | 31.66 | 27.80 | 48.27 |
> | SparseGPT | 55.25 | 37.77 | 77.45 | **73.68** | 75.25 | 43.27 | 31.80 | 56.35 |
> | Wanda     | 54.18 | **40.00** | 77.69 | 73.24 | 74.24 | 42.15 | 31.60 | 56.16 |
> | GBLM      | 54.68 | 37.35 | 75.24 | 73.12 | 74.68 | 42.32 | 30.80 | 55.46 |
> | Pruner-0  | 53.69 | 37.13 | 75.86 | 73.04 | 74.23 | 41.25 | 31.20 | 55.20 |
> | MaskPro   | **59.76** | 37.28 | **78.24** | 73.32 | **76.83** | **45.65** | **33.20** | **57.75** |
>
> **(8:16)-Sparsity of LLaMA-7B**
> |       | HellaS. | RACE | PIQA | WinoG. | ARC-E | ARC-C | OBQA | Avg. |
> | :-: | :-: | :-: | :-: | :-: | :-: | :-: | :-: | :-: |
> | Magnitude | 52.27 | 35.02 | 72.74 | 64.48 | 67.68 | 37.03 | 27.20 | 50.92 |
> | SparseGPT | 50.19 | 39.04 | 74.43 | 66.22 | 70.45 | 36.43 | 28.80 | 52.22 |
> | Wanda     | 49.77 | 39.14 | 75.30 | **66.61** | 70.62 | 36.18 | 28.80 | 52.35 |
> | GBLM      | 49.51 | **39.90** | 75.68 | 66.38 | 69.91 | 36.43 | 27.60 | 52.20 |
> | Pruner-0  | 50.12 | 38.68 | 75.22 | 66.13 | 69.93 | 35.48 | 27.80 | 51.91 |
> | MaskPro   | **53.15** | 39.23 | **76.15** | 66.56 | **72.87** | **40.13** | **29.60** | **53.96** |
>
> **(8:16)-Sparsity of LLaMA-13B**
> |       | HellaS. | RACE | PIQA | WinoG. | ARC-E | ARC-C | OBQA | Avg. |
> | :-: | :-: | :-: | :-: | :-: | :-: | :-: | :-: | :-: |
> | Magnitude | 55.43 | 37.51 | 74.48 | 66.06 | 68.94 | 38.05 | 27.60 | 52.58 |
> | SparseGPT | 54.24 | **40.38** | 77.15 | 70.19 | 75.08 | 41.31 | **31.00** | 55.62 |
> | Wanda     | 54.50 | 39.62 | 77.09 | 70.09 | 73.19 | 40.36 | 30.80 | 55.09 |
> | GBLM      | 54.45 | 39.18 | 76.35 | 69.92 | 73.75 | 40.07 | 29.60 | 54.76 |
> | Pruner-0  | 54.11 | 38.64 | 76.28 | 70.41 | 72.92 | 40.55 | 30.00 | 54.70 |
> | MaskPro   | **57.35** | 39.92 | **77.83** | **70.68** | **76.45** | **43.26** | 30.60 | **56.58** |
>
> We will include these training results in the final version of the paper. Notably, MaskPro remains effective when training 30B-scale models. Moreover, because of our linear probability modeling and the use of policy-gradient updates, MaskPro requires substantially fewer training resources while maintaining strong performance. We hope that these additional results address the reviewer's concerns regarding the effectiveness and scalability of our method. We are open to conducting further experiments based on any additional suggestions from the reviewers and will incorporate the findings into the final version of the paper.
>
> ## Q1: How sensitive is MaskPro's performance to the choice of $\alpha$?
> We thank the reviewer for raising this valuable question.
>
> In our PG update, the parameter $\alpha$ is used to track a stable estimate of the current baseline and prevent it from being overly influenced by the stochastic variance of sampled losses. Conceptually, this plays the same role as $\beta_1$ or $\beta_2$ in the Adam optimizer. To examine its sensitivity, we conducted the following set of experiments (the accuracy is the averaged accuracy of the 7 datasets above),
>
> | $\alpha$ | 0.7 | 0.9 | 0.95 | 0.99 | 0.995 |
> | :-: | :-: | :-: | :-: | :-: | :-: |
> | LLaMA-7B  | 34.25 | 49.28 | 49.37 | **49.62** | 49.21 |
> | LLaMA-13B | 38.68 | 52.23 | 52.78 | **53.02** | 52.74 |
>
> In our initial experiments, we found that using $\alpha=0.99$ consistently across all tasks provides the most stable and reliable performance. Therefore, we only report the selection of $0.99$ for reproduction. This hyperparameter requires almost no additional tuning.

---

> ### Author Response · Authors · 2025-11-22
> **Response to Reviewer NXWQ (2/2)**
>
> ## Q2: Does the sampling-without-replacement process still scale linearly in memory and remain computationally tractable for large M? This needs further discussion.
> We thank the reviewer for raising this insightful question.
>
> It is worth noting that, besides the standard sampling-without-replacement method, a more efficient exponential clocks method has been proposed in [1]. This method achieves sub-linear space complexity of $\mathcal{O}(M log(N/M))$ and simultaneously can avoid the need for updating the probabilities of remaining elements at each step of the sampling- without- replacement process, thus saving both computational time and space.
>
> The memory complexity is unavoidably linear which achieves the $\mathcal{O}(d)$ where $d$ is the model size. And the computational complexity is a particularly interesting process.
>
> We conducted the following tests at different scales and report the mask sampling time in each step on H100:
>
> |          | (2:4) | (4:8) | (8:16) |
> | :-: | :-: | :-: | :-: |
> | LLaMA-7B  | 2.328s | 1.334s | 0.794s |
> | LLaMA-13B | 4.739s | 2.692s | 1.574s |
>
> We use the torch.multinomial to sample masks within each group. We can observe that doubling the model size roughly doubles the sampling time. Another interesting observation is that the sampling time of (N:M)-Sparsity depends on $M$. With the same model size, a larger $M$ leads to shorter sampling time, due to parallel operation in the sampling processing. For a $d$-dimensional model, there are a total of $\frac{d}{M}$ sampling groups. Although increasing $N$ and $M$ makes each group more expensive to sample, the total number of parallel groups decreases proportionally. This reduction in the number of groups results in a more favorable computation pattern for hardware. Consequently, for more complex (N:M)-Sparsity, the time required for a single sampling step can actually be lower. **Large M is a GPU-friendly selection.**
>
> We once again thank the reviewer for the valuable suggestions and insightful questions. If there are any further concerns, we would be very happy to continue the discussion and address them.
>
> [1] Efraimidis, Pavlos S., and Paul G. Spirakis. "Weighted random sampling with a reservoir." Information processing letters 97.5 (2006): 181-185.

---

> > ### Comment · Reviewer_NXWQ · 2025-11-26
> > **Satisfied with the rebuttal**
> >
> > I am satisfied with the authors' rebuttal. The authors have addressed my concerns. I have raised my score.

---

> ### Author Response · Authors · 2025-11-26
> **Thank you again for your review**
>
> We sincerely appreciate the reviewers’ positive assessment of our rebuttal and are pleased to address correspondding concerns. We will further refine our manuscript based on the feedback provided in this round. Thank you once again for the time and effort you have dedicated to reviewing our submission!

---

### Author Response · Authors · 2025-11-24
**Summary of the Changes in the Updated Version**

We sincerely appreciate the reviewers for dedicating their valuable time and effort to evaluating our submission and for providing us with their insightful comments. During the rebuttal stage, we carefully addressed each reviewer's concerns and made our best effort to conduct broader empirical investigations, which were also the primary issues emphasized by most reviewers.

Here, we summarize the corresponding changes in the updated version, and most of these are based on additional empirical studies.

**Highlight and rethink the core contributions.**

We appreciate the professional suggestions provided by reviewer HgWG. In response, we have further highlighted the key contributions of this paper: (i) we establish a linear-space probabilistic relaxation to model the optimization problem under the strict N:M sparsity setting; and (ii) we introduce a reinforcement baseline method based on loss residuals. These enable the learning of a superior sparse mask while incurring training overhead comparable to rule-based approaches and substantially lower than that of existing training methods.

**Empirical studies of more sparsity**

We thank reviewers 4Q5R and rhLs for their valuable suggestions. In **Appendix A.7**, we have added the test results of the 7B and 13B models under the (8:16) sparsity setting. The consistently improved performance further demonstrates the effectiveness of our proposed method.

**Empirical studies of larger models**

We thank reviewers NXWQ, rhLs, and HgWG for their valuable suggestions. We have made our best effort to conduct training and evaluation on the 13B and 30B models, and the corresponding results have been added to **Appendix A.8**. Even on larger models, our proposed MaskPro method continues to deliver stable and consistent improvements in final test performance.

**Sensitivity studies of key hyperparameters**

We thank reviewers NXWQ, rhLs, and HgWG for their helpful suggestions. In **Appendix A.9, A.10, and A.11**, we provide sensitivity analyses of the tracker coefficient $\alpha$, the logits magnitude $C$, and the initialization of the mask on the 7B and 13B models. We have provided a detailed explanation of the core roles of the parameters $\alpha$ and $C$, and validated their effectiveness. **In addition, we thoroughly addressed a question commonly raised by the reviewers: random initialization can indeed be used for training, however, it substantially increases the required training duration in the RL-based methods. Therefore, we recommend starting from a simple initialization to accelerate the process.**

**Training with 1 sample**

We thank reviewer rhLs for the helpful suggestions on this point. In **Appendix A.12**, we report a comparison of results obtained when training with 1 sample versus 128 samples under different initialization settings, further demonstrating the effectiveness of the training process.

**Training time and acceleration**

We thank all four reviewers for raising the issue of the practical time cost of the proposed method in different scenarios, especially when scaling to larger models or more complex sparse states. In **Appendix A.13**, we report the time consumption and the proportion of time spent on each component during training for the 7B and 13B models. We also provide the sampling time under different coefficient patterns. **We further clarify that MaskPro incurs lower sampling time for complex sparse patterns, and its costs can reduce further as $M$ increases.**

Moreover, following the suggestion of reviewer rhLs and HgWG, we also provide a comparison of the acceleration achieved by several existing approximate sampling schemes and their corresponding performance degradation **in Appendix A.14**. This provides empirical support for the further acceleration potential of MaskPro.

**We once again thank all reviewers for the time and constructive feedback. We have incorporated the relevant discussions into the revised version. We look highly forward to your further feedback.**

---

### Meta-Review · Area_Chair_wuUA · 2026-01-04

**Summary:**

The key reviewer concerns that informed the decision can be summarized as follows:

1. Reviewers questioned whether MaskPro generalizes beyond 2:4 sparsity and 7B-scale models, particularly to larger models (13B/30B) and more complex sparsity patterns such as 8:16.

2. There were concerns about the stability of the policy-gradient optimization, including sensitivity to the tracker coefficient, logits magnitude, and mask initialization.

3. Reviewers requested clearer evidence regarding training time, memory overhead, and sampling cost, especially compared to rule-based and existing learning-based sparsity methods.

4. Questions were raised about whether the sampling-without-replacement procedure remains tractable and memory-efficient as model size and M increase.

Overall, these concerns focused on empirical completeness and robustness rather than questioning the core idea or correctness.

**Reviewer Concerns:**

Concerns addressed by the rebuttal:

1. The authors added extensive new experiments on 13B and 30B models, as well as on 8:16 sparsity. These results consistently show that MaskPro maintains strong performance and validates the scalability of the linear-space probabilistic formulation.

2. Detailed sensitivity studies were provided for key parameters (tracker coefficient, logits magnitude, initialization), demonstrating stable performance across a wide range of values and supporting claims of robustness.

3. The rebuttal includes training time breakdowns, memory usage, and sampling time profiles, clearly showing that MaskPro achieves substantially lower overhead than prior learning-based approaches and comparable efficiency to rule-based methods.

4. The authors provided both theoretical discussion and empirical timing results, including references to alternative sampling strategies, convincingly showing that the sampling process scales linearly in memory and remains computationally tractable.

Concerns still outstanding (non-blocking)

1. While results on 30B models are provided, experiments on 70B-scale models are not included due to resource constraints. This limitation is clearly acknowledged and is reasonable given the scope and cost of such experiments.

2. Some reviewers suggested exploring joint optimization of sparsity masks and model weights. This is a natural future direction rather than a requirement for acceptance.

**Reviewer Scores:**

Reviewer NXWQ (Initial rating: 6)
Likely raise to 8. The reviewer explicitly stated satisfaction with the rebuttal and raised their score after additional experiments and clarifications.

Reviewer 4Q5R (Initial rating: 8)
Likely to remain at 8. The rebuttal directly addressed concerns about larger sparsity patterns, reinforcing an already positive assessment.

Reviewer HgWG (Initial rating: 6 )
Likely to  remain unchanged. The reviewer indicated support for acceptance without changing rating.

Reviewer rhLs (initial rating 2)
The rebuttal addressed the reviewer’s requests for additional large-scale experiments and training analysis. While still possibly skeptical, the main technical concerns were addressed.

Overall, post-discussion sentiment trends clearly upward, with the majority of reviewers supportive of acceptance.

---

### Decision · Program_Chairs · 2026-01-26

Accept (Poster)